# Membrane to cortex attachment determines different mechanical phenotypes in LGR5+ and LGR5- colorectal cancer cells

Sefora Conti[1], Valeria Venturini [2], Adrià Cañellas-Socias [3,4], Carme Cortina[3,4], Juan F. Abenza[1], Camille Stephan-Otto Attolini [3], Emily Middendorp Guerra[3,4], Catherine K. Xu [5], Jia Hui Li[6], Leone Rossetti[1], Giorgio Stassi [7], Pere Roca-Cusachs [1,8], Alba Diz-Muñoz[6], Verena Ruprecht [2,9,10], Jochen Guck [5,11,12], Eduard Battle [3,4,10] ✉, Anna Labernadie [1,13] ✉ & Xavier Trepat [1,8,10,14] ✉

Colorectal cancer (CRC) tumors are composed of heterogeneous and plastic cell populations, including a pool of cancer stem cells that express LGR5. Whether these distinct cell populations display different mechanical properties, and how these properties might contribute to metastasis is poorly understood. Using CRC patient derived organoids (PDOs), we find that compared to LGR5- cells, LGR5+ cancer stem cells are stiffer, adhere better to the extracellular matrix (ECM), move slower both as single cells and clusters, display higher nuclear YAP, show a higher survival rate in response to mechanical confinement, and form larger transendothelial gaps. These differences are largely explained by the downregulation of the membrane to cortex attachment proteins Ezrin/Radixin/Moesin (ERMs) in the LGR5+ cells. By analyzing single cell RNA-sequencing (scRNA-seq) expression patterns from a patient cohort, we show that this downregulation is a robust signature of colorectal tumors. Our results show that LGR5- cells display a mechanically dynamic phenotype suitable for dissemination from the primary tumor whereas LGR5+ cells display a mechanically stable and resilient phenotype suitable for extravasation and metastatic growth.

The intestinal epithelium is the fastest self-renewing tissue in our body[1,2]. Intestinal stem cells (ISC) fuel this renewal by dividing into transit amplifying cells which in turn differentiate into multiple lineages of short-lived specialized cells[2]. In colorectal cancer (CRC), essential features of this hierarchical structure are maintained, resulting in intratumor heterogeneity[2]. Among the heterogeneous cell populations composing colorectal adenomas and carcinomas, a subset of cells called cancer stem cells (CSC) has a high tumor-initiating ability and expresses a genetic signature similar to the ISCs[3–5]. Leucine-rich repeat-containing G-protein coupled receptor 5 (LGR5) is a reliable marker for adult ISCs and was proven to mark functional CSCs in

CRC[6–9]. Recent lineage tracing and selective cell ablation experiments have confirmed the essential role of LGR5+ CSCs in tumor and metastatic growth. Nevertheless, they also revealed the importance of transitions between differentiation states as an adaptive mechanism to ensure cancer progression and endure changing microenvironmental pressures[10–13]. In mouse CRC models, genetic ablation of LGR5+ CSCs that had colonized the liver inhibited metastasis formation[10]. However, it was found that most metastases were initially seeded by LGR5- cells; only after metastatic colonization, conversion of LGR5- to LGR5+ enabled metastatic growth[11]. Further reinforcing this model, we recently showed that a subset of LGR5- tumor cells (named high relapse

---

cells · HRCs) that remain hidden in foreign organs after surgical removal of the primary CRC are responsible for metastatic relapse[14]. Altogether, these studies highlight the importance of microenvironmental stimuli in shaping the role of CSCs in tumor growth and progression.

Metastasis formation is a sequential process involving multiple steps including cancer cell dissemination, intravasation, survival in the bloodstream, adhesion to the vessel wall, extravasation, and colonization of distant organs[15]. In every step of this cascade, distinct mechanical properties may contribute to the survival and metastatic potential of cancer cells[16,17]. For instance, reduced cell stiffness contributes to the ability of cancer cells to migrate through confining microenvironments[18], while it can be detrimental for the survival of circulating tumor cells (CTCs) against hemodynamic stresses in the blood stream[19]. Another mechanical property that affects cell metastatic potential is adhesion. Fine-tuning the balance between cell-cell and cell-ECM adhesions is required all throughout the metastatic cascade and enables motility, cell-cell interactions, and integration of mechanical stimuli from the environment[20]. Several studies have related malignant transformation to changes in mechanical phenotypes including decreased cell stiffness[18,21,22], as well as changes in cell shape[23,24], cell-cell[25–29] and cell-ECM[30–33] adhesions. Nevertheless, the role of cellular mechanical properties in intra-tumor heterogeneity and its functional implications in CRC metastasis remains largely unexplored. In this study, we show that LGR5+ and LGR5- cells from patient-derived CRC organoids exhibit distinct mechanical phenotypes in terms of stiffness, adhesion, migration, YAP localization, response to confinement, and transendothelial migration. We show that these responses are mediated by differential expression of Ezrin/Radixin/ Moesin (ERM) proteins, responsible for tethering the plasma membrane to the underlying actin cytoskeleton. The distinct mechanical phenotypes of LGR5+ and LGR5- cells are suitable for different functions during CRC metastasis formation.

## Results

### LGR5+ and LGR5- cells display differences in morphology, adhesion, stiffness and YAP localization

Patient-derived Organoids (PDOs) are 3D structures, derived from biopsies, which self-organize and retain essential features of the in vivo tumor in terms of architecture, heterogeneity and gene expression profile[10,34–36]. To mechanically characterize LGR5+ and LGR5- cells we used a previously established PDO model that carries genetic alterations in four main pathways driving colorectal carcinogenesis (WNT, EGFR, TGF-β and p53, see methods)[35]. The PDOs were engineered by knock-in CRISPR/Cas9-mediated homologous recombination where an IRES-iCasp9-T2A-TdTomato-WPRE-BGHpolyA cassette was inserted after the stop codon of the *Lgr5* gene, thereby fluorescently labeling the LGR5+ cells using the LGR5 endogenous promoter as a driver of TdTomato expression[37]. The organoids were able to differentiate in vitro under normal culture conditions giving rise to heterogeneous populations expressing the stem-cell marker LGR5 and the differentiation marker cytokeratin 20 (CK20), mainly in a mutually exclusive manner (Fig. 1a, Supplementary Fig. 1a and Supplementary Movie 1). We sorted the cells based on LGR5 expression into 3 groups expressing high (LGR5+), medium (LGR5med), or low (LGR5-) levels of Tdtomato (Fig. 1b and Supplementary Fig. 1b, c) and characterized them through bulk RNAseq. Differential expression analysis revealed that the LGR5+ population was enriched in gene signatures corresponding to ISCs, proliferation, biosynthesis[37] and resistance to chemotherapy[38] (Supplementary Fig 1d). Conversely LGR5- cells showed enrichment in gene signatures associated to intestinal differentiation (enterocyte, goblet, tuft and mucus-secreting cells) as well as fetal state signatures. LGR5- cells also showed an upregulation of markers for high relapse cells (HRCs)[14], including the expression of *Emp1* (encoding epithelial membrane protein 1) (Supplementary Fig. 1d, f). We further

investigated the differential expression of Gene Ontology Biological Process (GOBP) gene sets related to cell shape, migration and structural changes in the three populations. In the LGR5+ cells we found enrichment in genes associated with adhesion-dependent cell spreading and EMT. Conversely, in LGR5- cells we found enrichment in genes associated with epithelial migration, cytoskeleton and plasma membrane organization (Supplementary Fig. 1e, Supplementary Table 1, Supplementary Data 1).

This transcriptomic analysis suggests that single cells within the organoids might display differences in mechanical phenotype as a function of their LGR5 expression. To test this possibility, we dissociated single cells from the organoids and seeded them on collagen I-coated polyacrylamide (PAA) gels with a stiffness spanning the physiological and pathological range for colon[39] (Fig. 1b). The three groups showed clear differences in cell shape. LGR5- cells were rounder and less adhesive whereas LGR5+ spread on the substrate and adopted an elongated morphology (Fig. 1c, d, Supplementary Fig. 1c). These differences were conserved across the probed stiffness range (Fig. 1c). Over the course of 100 h in culture medium promoting stemness, sorted single cells from both LGR5- and LGR5+ populations displayed an increase in Tdtomato fluorescence. This expected increase was paralleled by the acquisition of a more elongated morphology (Supplementary Fig. 1g–i), further supporting a coupling between cell shape and LGR5 expression.

To explore whether differences in roundness coincided with changes in mechanotransduction, we stained for the nuclear effector of the Hippo pathway Yes-associated protein (YAP), which not only regulates intestinal development[40], regeneration[41] and tumorigenesis[41,42], but is also a well-established mechano-sensor[43]. Consistent with the trends of cell spreading, YAP nuclear localization was highest in the LGR5+ cells, decreased in the intermediate group, and was lowest in the LGR5- cells (Fig. 1e, f). This localization pattern was maintained in all tested rigidities from soft (0.5kPa) to stiff (30kPa), revealing a lack of response to stiffness in YAP localization of both LGR5+ and LGR5- cells.

To determine whether the differences in shape and spreading were coupled to distinct migratory behaviors we tracked the cell movement on the PAA gel substrates. No differences in velocities of LGR5- and LGR5+ single cells were found in all tested stiffnesses (Fig. 1g). Next, we used traction force microscopy (TFM) to investigate the relation between CRC differentiation states and the ability to generate traction forces. We found no differences in the mean traction exerted by LGR5-, LGR5med and LGR5+ cells (Fig. 1h). Nevertheless, the morphological differences observed indicated a possible difference in the spatial force pattern. To explore this, we decomposed the traction stress field into a simpler quantity called the dipole moment. This quantity represents the rotational asymmetry of the stress field and defines the major and minor axis of elongation of the cell[44]. Thus, we used the ratio between the major and minor dipole (Mδ) to quantitatively assess the asymmetry of force distribution in the LGR5-, LGR5med and LGR5+ cells. Compared to the LGR5- cells, the two positive groups displayed higher Mδ, indicating a more anisotropic mechanical state (Fig. 1i, j).

We next investigated cell stiffness as a function of their stemness. To do so, we used real time deformability cytometry (RT-DC), a high-throughput microfluidic technique where cells suspended in a highly viscous medium are flowed through a narrow channel and deformed by shear stress[45,46] (Fig. 2a). We found that, compared to the LGR5med and LGR5- cells, LGR5+ cells were less deformed upon application of this shear stress, indicative of a higher stiffness (Fig. 2b, c). Accordingly, they displayed a higher apparent elastic modulus (Fig. 2d).

### LGR5+ cells respond to confinement with higher survival and slower ameboid migration

Throughout the various steps of the metastatic cascade, cancer cells can encounter confining microenvironments including heterogeneous

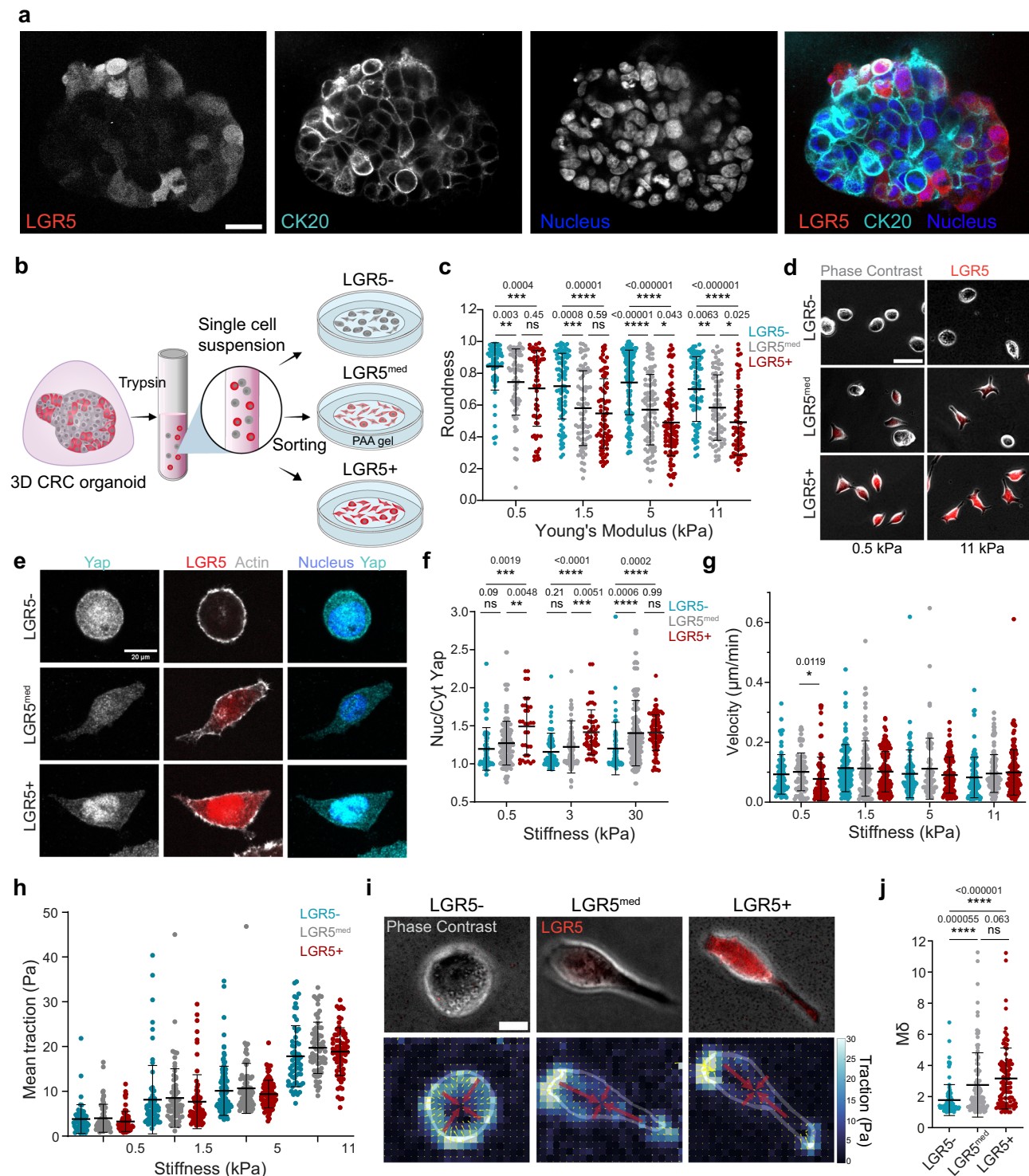

ECM geometries, 3D longitudinal tracks formed from aligned collagen bundles, porous matrices and narrow inter-endothelial spaces[47]. Physical confinement and low adhesion have been shown to promote a fast bleb-based amoeboid-like migration in mesenchymal cells and immune cells, as well as in transformed cells[48–51]. Having observed stemness dependent variations in cell spreading, force generation and stiffness, we hypothesized that LGR5+ cells may respond differently to confinement compared to LGR5- cells. To test this hypothesis, we used a dynamic cell confiner[52] (Fig. 2e). Single cells (LGR5- and LGR5+ cells mixed at 1:1 ratio) were confined to a precise height between two low-adhesion parallel surfaces. In the absence of confinement and adhesion (suspended cells), the vast majority of PDO cells displayed a round

morphology with no blebs (Supplementary Fig. 1k) and an average diameter of $9.3 \pm 1.2\,\mu m$. As we increased the levels of confinement (7 and $4.5\,\mu m$ pillar height), the fraction of blebbing cells grew and we identified four distinct response categories: no blebbing, round non-polarized blebs, elongated polarized blebs and cell death (Fig. 2f, Supplementary Fig. 1k). Analysis of the distribution of the cell responses at $4.5\,\mu m$ confinement revealed a higher tendency of the LGR5+ cells to adopt an amoeboid-like blebbing behavior compared to LGR5- cells (Fig. 2g, Supplementary Movies 2, 3). Notably, the LGR5- cells showed a substantial increase in cell death in response to confinement (Fig. 2g). Next, we investigated whether the response to confinement also involved changes in cell migration in each of the four

**Fig. 1 | LGR5+ and LGR5- single cells display differences in adhesion, polarity and YAP localization. a** CRC PDOs cultured for 1 week in culture matrix gel stained for CK20 and nuclei. LGR5+ cells are labeled with Tdtomato. Scale bar, 20 μm. Representative images from two independent experiments. **b** Scheme illustrating the preparation of PDOs for single cell analysis on 2D soft substrates. **c** Cell roundness measured for sorted cells seeded on collagen-I coated gel substrates of 0.5, 3, 5, 11 kPa in stiffness. **d** LGR5-, LGR5^med and LGR5+ cells on 0.5 and 11 kPa gel substrates. Representative images of four independent experiments. Scale bar, 50 μm. **e** Single PDO cells seeded on 0.5 kPa gel substrates stained for Actin, nuclei and YAP. LGR5+ cells are labeled with Tdtomato. Representative images of four independent experiments. Scale bar, 10 μm. **f** Quantification of YAP nuclear/cytoplasmic ratio of PDO cells seeded on 0.5, 3, 30 kPa gels. In (**c**) and (**f**) $n \geq 60$ cells/condition from four independent experiments. Statistical significance was determined using two-way analysis of variance, followed by a Šidák multiple-comparison test. Quantification of cell velocity (**g**) and mean

traction (**h**) exerted by sorted cells on gels of 0.5, 1.5, 5 and 11 kPa in stiffness. In (**g**) and (**h**) data are represented as the mean ± s.d. of $n \geq 60$ cells/condition from four independent experiments. Statistical significance was determined using Shapiro-Wilk normality test, followed by a Kruskal-Wallis multiple-comparison test. **i** LGR5-, LGR5^med and LGR5+ cells seeded on 3 kPa gels and their corresponding traction stress field and force dipole. The traction stress vectors and their amplitude are represented by the yellow arrows and colormap, respectively. The two eigenvectors of the dipole matrix are represented by red arrows. Representative images of four independent experiments. Scale bar, 10 μm. **j** Quantification of the polarization Mδ of LGR5-, LGR5^med and LGR5+ cells. $n \geq 95$ cells/condition from four independent experiments. Statistical significance was determined using Shapiro-Wilk normality test, followed by a Kruskal–Wallis multiple-comparison test. All data are represented as the mean ± s.d. Source data are provided as a Source Data file.

categories. LGR5- cells displaying polarized blebs moved significantly faster than LGR5+ cells in the same response category, while no significant differences were observed in the migration speed of other categories (Fig. 2h, i, Supplementary Fig 1l, Supplementary Movies 4, 5). These experiments show that LGR5- cells are more fragile under confinement (higher death rate), but those that survive migrate faster.

## LGR5- clusters are rounder, migrate faster and exert less traction

Metastases can be seeded not only by single cells but also by highly cohesive epithelial cell groups[53-57]. We thus explored whether multicellular clusters displayed a mechanical phenotype similar to single cells. To study the mechanics of CRC clusters we extracted self-assembled PDOs from 3D matrix gels and seeded them on gel substrates coated with collagen I (Fig. 3a). We then studied the mechanical properties of cell clusters dividing them into 3 groups based on LGR5 expression (high, medium and low TdTomato cell clusters), in analogy with our single cell analysis (Fig. 3b, c). Similarly to single cells, clusters expressing low levels of LGR5 displayed higher roundness on collagen I coated substrates (Fig. 3b–d). However, unlike single cells, their migration speed was higher and their mean traction forces were lower (Fig. 3e, f and Supplementary Movies 6, 7). The same trend in cluster speed was observed on a stiffer substrate (Supplementary Fig. 2a, b). Quantification of myosin light chain phosphorylation (pMLC) of the cluster's most basal plane indicated higher levels of acto-myosin contractility in the LGR5^high clusters compared to LGR5^low (Supplementary Fig. 2c–e).

These results suggest that cluster spreading and migration can be understood within the biophysical framework of active wetting[58]. Inspired by how a fluid droplet wets a surface, this framework provides a biophysical understanding of how an active contractile cell aggregate spreads and moves on a substrate in terms of a balance between in-plane cell-substrate traction, out-of-plane surface tension and tissue contractility[58-62]. A feature of active wetting is that cluster migration is a non-monotonic function of the cell-substrate contact angle; for either high or low spreading (low and high contact angles, respectively) clusters move slowly, but at intermediate spreading (neutral wetting, ~90° contact angles) clusters move rapidly[63]. To test whether the active wetting framework captures our experiments, we measured the contact angle of LGR5^high, LGR5^med and LGR5^low populations. We found that contact angles decreased with LGR5 expression and, remarkably, that the fastest population (LGR5^low) displayed contact angles close to neutral wetting regime, where velocity is predicted to be maximal (Fig. 3g–i). These data support that cluster migration can be explained in terms of active wetting. They highlight, further, that multicellular clusters retain features of single cell behavior but also display emergent mechanical properties which are absent at the single cell level.

## LGR5+ clusters adhere better to the endothelium and form transendothelial gaps

One of the fundamental steps in the metastatic cascade is cancer cell extravasation. Circulating tumor cells are found both as single cells and clusters, the latter being less frequent but with higher metastatic potential[57]. Given the differences in cluster morphology and spreading observed on collagen substrates, we investigated whether clusters with distinct levels of stemness displayed different capacities to adhere to and breach endothelial monolayers. We thus grew cohesive human umbilical vein endothelial cells (HUVEC) monolayers on collagen-I substrates and confirmed the presence of stable and mature cell-cell junctions through immunostaining for VE-Cadherin (Supplementary Fig. 2f). We then seeded PDOs on top of the monolayer and started imaging one hour after seeding (Fig. 4a). We observed that some of the clusters attached to the endothelial cells and created a gap through the monolayer while other clusters detached and floated away from the field of view (Fig. 4b and Supplementary Movies 8–10). This behavior was independent of the cluster cross-sectional area (Supplementary Fig. 2g). Clusters that remained attached during the whole timelapse exhibited higher mean Tdtomato fluorescence compared to clusters that detached (Fig. 4c). Furthermore, we quantified the percentage of clusters able to form a gap in the endothelial monolayer as a function of LGR5 expression. Within the three cluster groups (Supplementary Fig. 2h), clusters containing more LGR5+ cells were more competent in forming a gap (Fig. 4d) as indicated by the higher percentage of gap forming clusters, the shorter time needed for gap formation and the higher gap area (Fig. 4e, f).

## ERM proteins determine the mechanical differences between LGR5+ and LGR5- cells

We next sought to identify the molecular mechanisms that explain the distinct mechanical phenotypes of LGR5+ and LGR5- cells and clusters. The differences in adhesion and blebbing in response to confinement suggest a role for tethering between the cell membrane and the cytoskeleton, which is mainly governed by Ezrin, Radixin and Moesin, collectively known as ERM proteins[64,65]. Indeed, overexpression of these proteins has been shown to induce cellular rounding in adherent single cells[66-68] and to inhibit bleb formation and protrusion[69-71]. Hence, we investigated whether the identified phenotypes in PDO cells are linked to a differential expression of the ERM proteins. RNAseq data identified the downregulation of gene signatures associated with membrane to cortex attachment in LGR5+ cells (Fig. 5a). RT-qPCR confirmed these results, revealing a significant downregulation of each of the three ERMs in LGR5+ cells (Fig. 5b), which was further confirmed at the protein level (Fig. 5c).The membrane to cortex attachment (MCA) activity of ERM proteins requires a two-step activation process that involves phosphorylation of a conserved threonine residue present in the F-actin

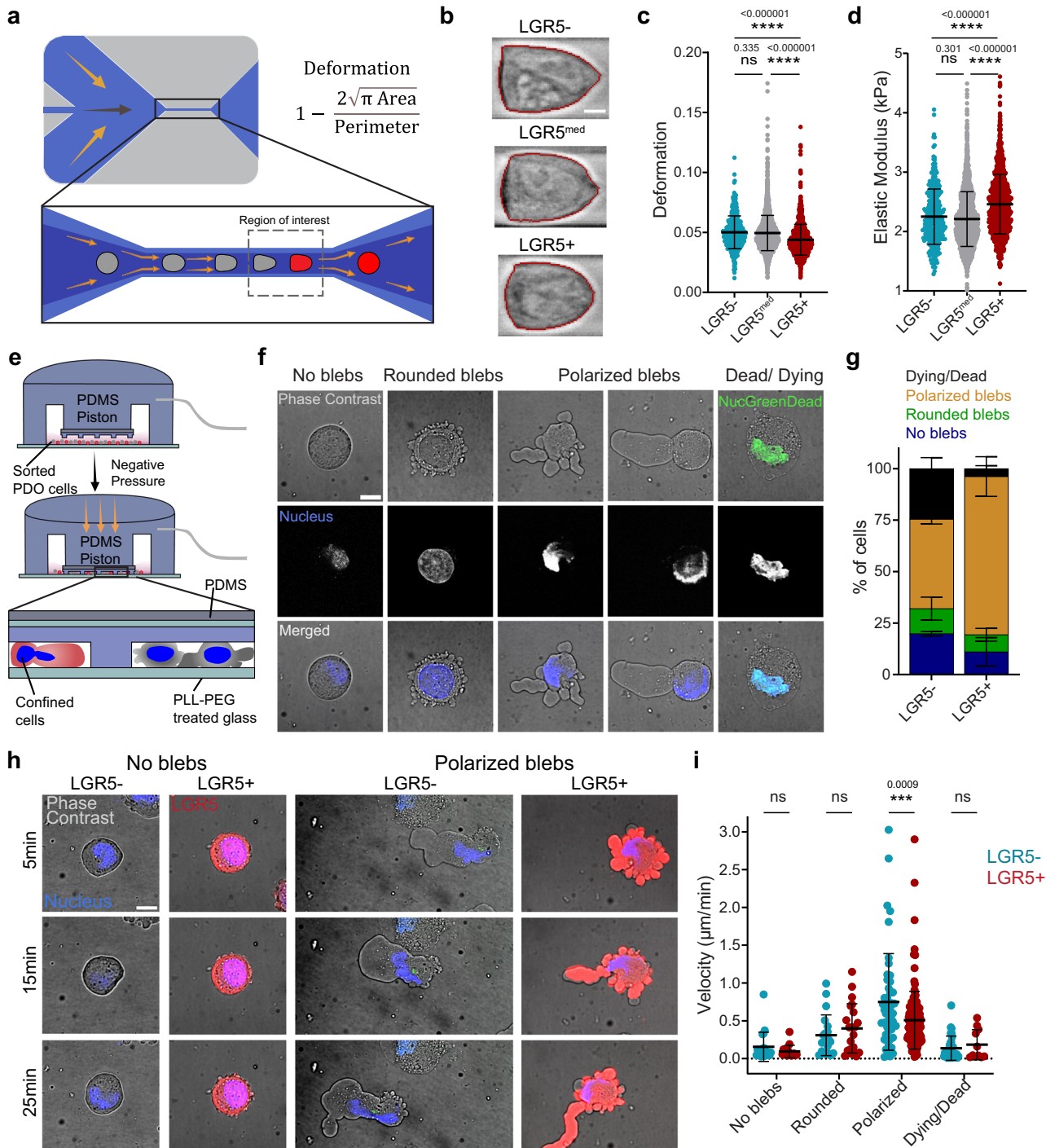

**Fig. 2 | LGR5+ are stiffer and respond to confinement with a higher survival rate and slower ameboid migration. a** Scheme of the real time-deformability cytometry (RT-DC) setup. Cells suspended in a highly viscous solution are flushed through a microfluidic channel of 20 μm height and width, where they are deformed by shear stress and pressure gradients. Cells are imaged in the region of interest (dashed rectangle) to determine their degree of deformation. **b** Representative images of LGR5-, LGR5med and LGR5+ analyzed with RT-DC. Scale bar, 5 μm. **c**, **d** Cell deformation and derived elastic modulus of LGR5-, LGR5med and LGR5+ cells. Data are represented as the mean ± s.d. from one experiment representative of four independent ones. $n \geq 300$ cells/subgroup. Statistical significance was determined using Shapiro-Wilk normality test, followed by a Kruskal–Wallis multiple-comparison test. **e** Scheme of dynamic cell confiner. The central PDMS piston holds a glass slide to which PDMS pillars of fixed height (4.5 μm) are attached. The device acts as a suction cup. When negative pressure is applied, the central piston is pressed down, thereby confining the cells. **f** Representative

examples of the four response categories identified in confined LGR5- and LGR5+ cells. Scale bar, 10 μm. **g** Percentage of dying/dead, polarized blebs, rounded blebs and no blebs in LGR5- and LGR5+ cells under confinement on a non-adhesive surface. Data are represented as the mean ± s.d. of percentages of $n \geq 112$ cells/subgroup from four independent experiments. Statistical significance was determined using two-way analysis of variance, followed by a Šidák multiple-comparison test. **h** Representative time lapse images of LGR5- and LGR5+ no blebs, LGR5+ and LGR5- polarized blebs. Scale bar, 10 μm. In (**f**, **h**) images are representative images of four independent experiments, with a total of 303 cells. **i** Migration velocity of tracked nuclei of LGR5- and LGR5+ cells, divided into categories according to the confinement response. Data are represented as the mean ± s.d. of $n \geq 112$ cells/subgroup from four independent experiments. Statistical significance was determined using two-way analysis of variance, followed by a Šidák multiple-comparison test. Source data are provided as a Source Data file.

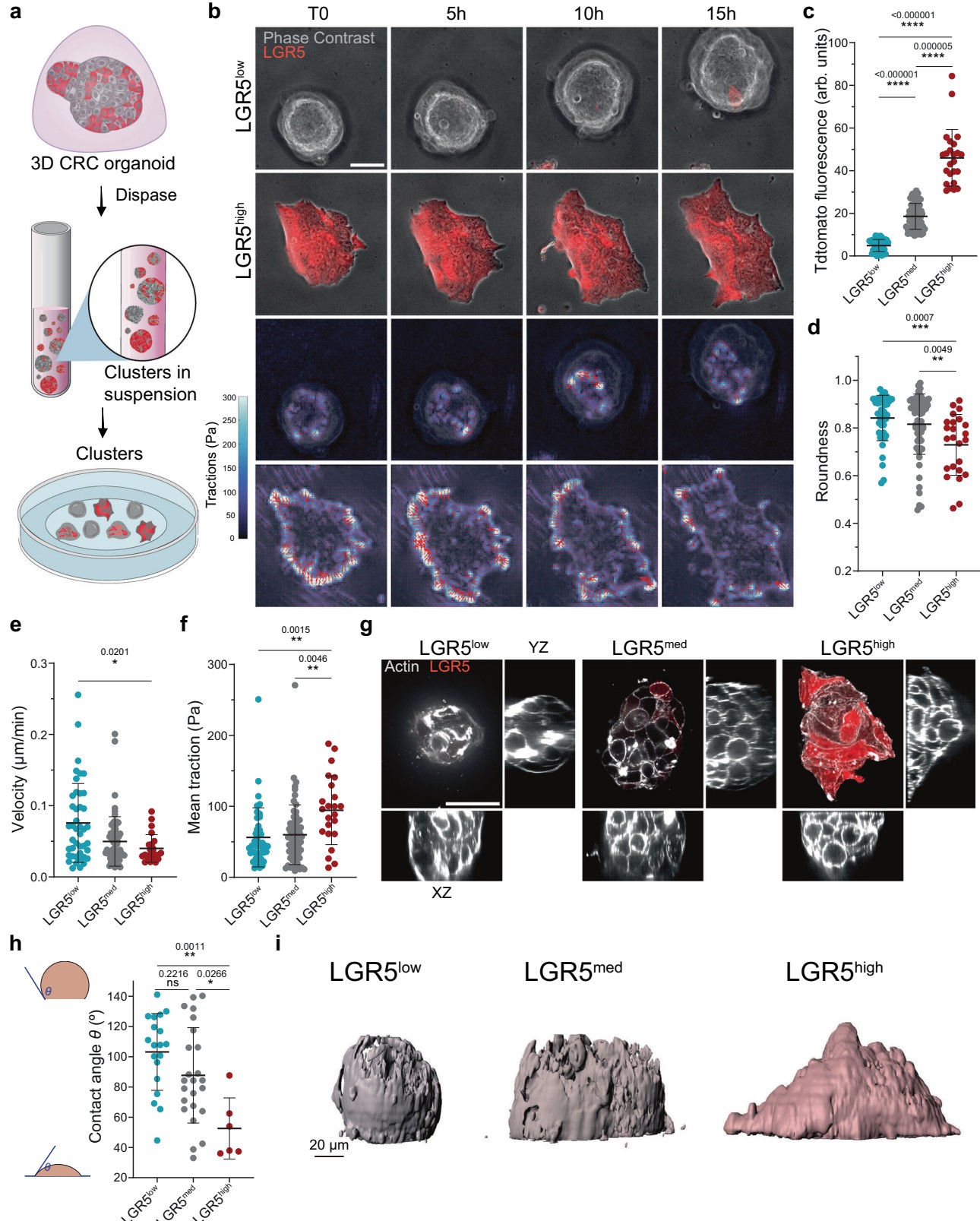

binding site[65,72,73]. Measurement of the phosphorylated form of ERM proteins confirmed their increased activity in the LGR5- cells (Fig. 5c).

To investigate the link between ERM upregulation and the mechanical phenotype of LGR5- cells we created a stable PDO line expressing shRNAs which target Ezrin, Radixin and Moesin (Supplementary Table 2). This line displayed an efficient silencing of Radixin and Moesin and mild downregulation of Ezrin (Fig. 5e). The obtained overall ERM downregulation led LGR5- cells to exhibit a more elongated shape, with a mean cell roundness value similar to the LGR5med cells (Fig. 5f, g). Moreover, ERM downregulation induced in the LGR5low clusters a mechanical phenotype similar to the LGR5high clusters, as indicated by higher spreading and by the slower migration on soft substrates (Fig. 5j, k).

**Fig. 3 | LGR5- clusters are rounder, migrate faster and exert less traction.**
**a** Scheme of the preparation of PDOs for cluster analysis on 3kPa 2D substrates.
**b** Time lapse of LGR5[low] and LGR5[high] clusters (first and second rows, respectively).
Bottom rows show the time lapse of the traction stress field of the clusters shown in
the top rows. Representative images of $n \geq 24$ clusters/ subgroup from four inde-
pendent experiments. Scale bars, 50 µm. Tdtomato fluorescence intensity (**c**),
cluster roundness (**d**), migration speed (**e**), and mean traction (**f**). Data are repre-
sented as the mean ± s.d. of $n = 128$ clusters from four independent experiments.
Statistical significance was determined using Shapiro-Wilk normality test, followed

by a Kruskal−Wallis multiple-comparison test. **g** LGR5[low], LGR5[med] and LGR5[high]
clusters labeled with Sir-Actin and their corresponding XZ and YZ lateral planes.
Scale bars, 50 µm. Representative images of two independent experiments. $n = 49$
clusters. **h** Mean contact angle $\theta$ between the cell cluster and the 3 kPa substrate in
LGR5[low], LGR5[med] and LGR5[high] clusters. Data are represented as the mean ± s.d. of
$n \geq 6$ clusters/ subgroup from two independent experiments. Statistical sig-
nificance was determined using one-way analysis of variance, followed by a Šidák
multiple-comparison test. **i** 3D rendering of the same clusters in (**g**). Source data are
provided as a Source Data file.

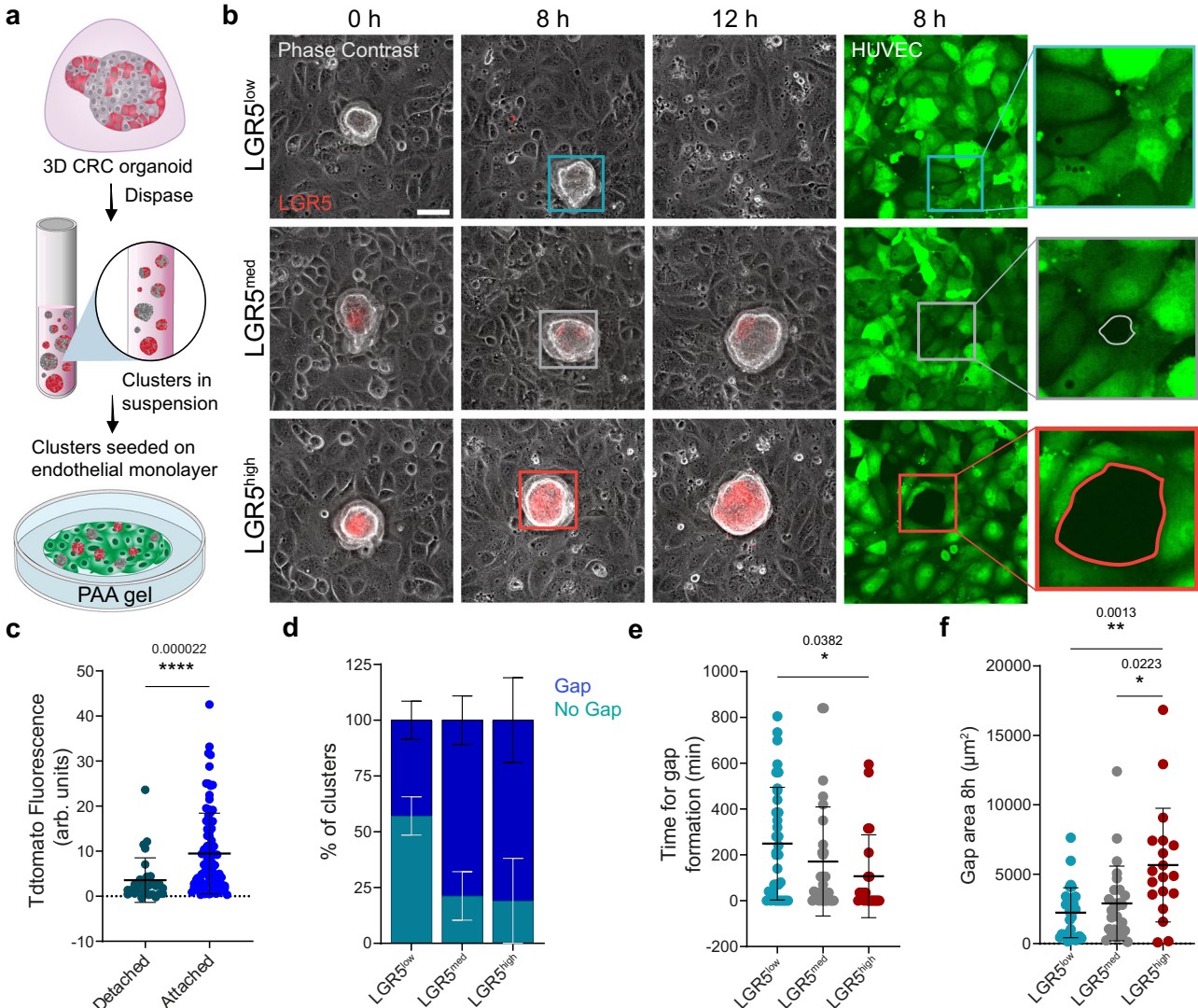

**Fig. 4 | LGR5+ adhere better to endothelial monolayers and form
transendothelial gaps. a** Schematic of the preparation of PDOs for adhesion
analysis on an endothelial monolayer. **b** PDO clusters seeded on HUVEC mono-
layers formed on collagen I coated 3 kPa gel. Representative time lapse of LGR5[low],
LGR5[med] and LGR5[high] clusters on a HUVEC monolayer. The last two columns show
the fluorescence image of the HUVEC monolayers at 8 h including a zoom at the
contact point between clusters and monolayers. Scale bars, 50 µm. Representative
images from four independent experiments. **c** Mean Tdtomato fluorescence
labeling LGR5+ cells in CRC clusters that either remained attached to the monolayer
during the whole time-lapse or detached. Data are represented as the mean ± s.d. of
$n \geq 31$ clusters/ subgroup clusters from four independent experiments. Statistical
significance was determined using Shapiro-Wilk normality test, followed by a

Kruskal−Wallis multiple-comparison test. **d** Percentage of clusters that formed a
gap in the endothelial monolayer for LGR5[low], LGR5[med] and LGR5[high] cell clusters.
Data are represented as the mean ± s.d. of percentages from four independent
experiments. Statistical significance was determined using two-way analysis of
variance, followed by a Šidák multiple-comparison test. **e** Time for gap formation
for LGR5[low], LGR5[med] and LGR5[high] cell clusters. **f** Gap area measured 8 h after
beginning of time lapse acquisition. In (**e**) and (**f**) data are represented as the
mean ± s.d. of $n = 77$ clusters that remained attached during time lapse acquisition
from four independent experiments. Statistical significance was determined using
Shapiro-Wilk normality test, followed by a Kruskal−Wallis multiple-comparison
test. Source data are provided as a Source Data file.

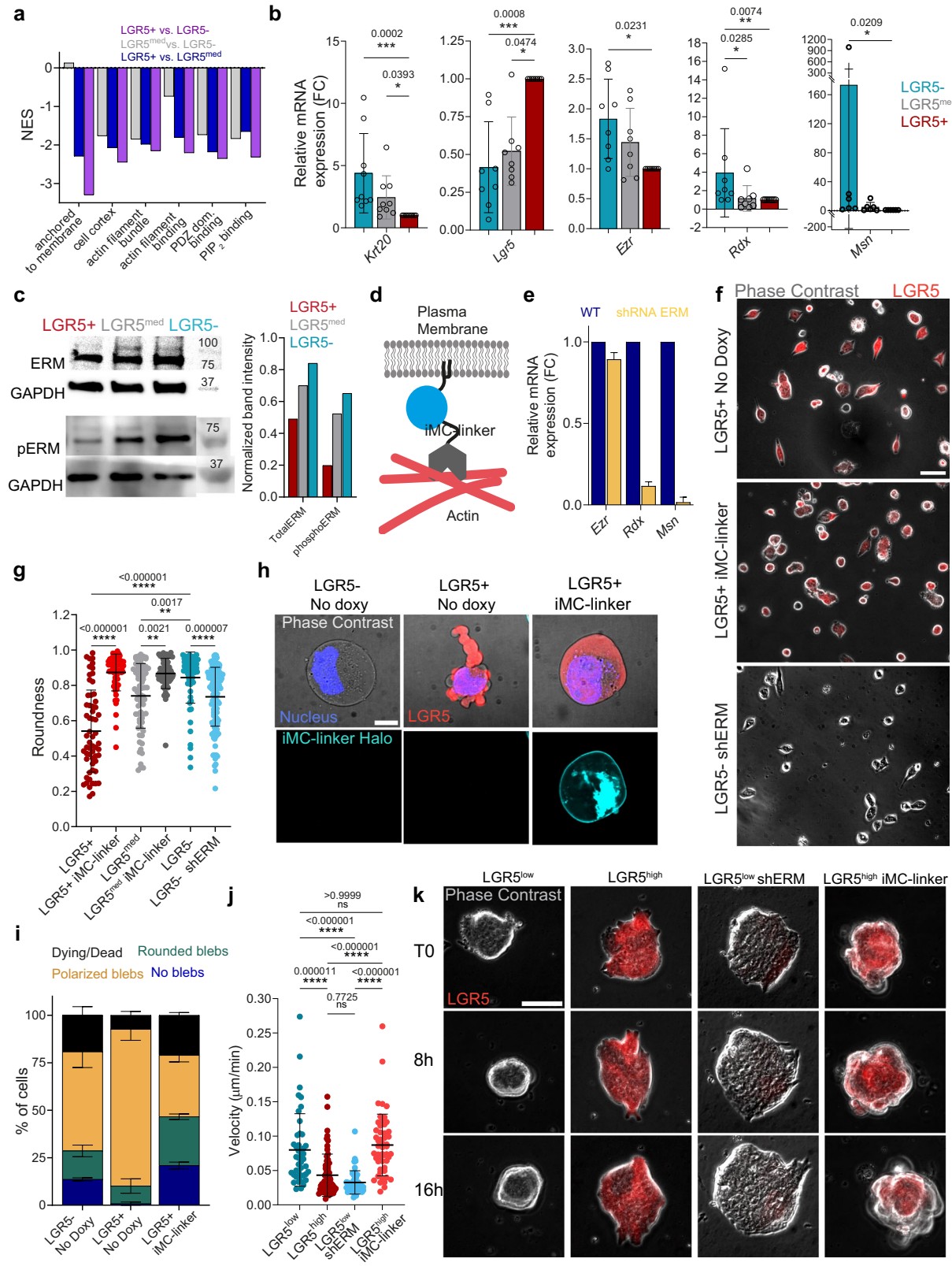

To further test whether differences in MCA are indeed responsible for the differences between LGR5+ and LGR5- phenotypes, we used a previously developed inducible synthetic linker[67] (iMC-linker) that mimics the ERM tethering activity (Fig. 5d), without modulating the downstream signaling pathways associated with ERM activation. Induction of the iMC-linker by doxycycline treatment in the LGR5+ and

LGR5[med] cells resulted in a markedly rounder shape (Fig. 5f, g) similar to that observed in LGR5- cells.

We hypothesized that the higher MCA in the LGR5- cells could account for their lower propensity to adopt a blebbing phenotype under confinement. To test this hypothesis, we induced expression of the iMC-linker in the LGR5+ cells and confined them at 4.5 µm.

**Fig. 5 | ERM proteins determine the mechanical differences between LGR5+ and LGR5- cells. a** Normalized Enrichment Scores (NES) from enrichment analysis ROAST-GSA for selected GO cellular component (GOCC) gene sets in LGR5-, LGR5$^{med}$ and LGR5+ cells. *p* values are listed in Supplementary Table 1. Data obtained from two independent replicates (at least 20,000 cells per condition). **b** Relative expression of *Lgr5*, *Krt20* and ERM proteins for sorted cells. Data are represented as the mean ± s.d. from nine independent experiments. Statistical significance was determined using Shapiro-Wilk normality test, followed by a Kruskal−Wallis multiple-comparison test. **c** Levels of ERM and phospho-ERM proteins in sorted PDO cells with quantification of band intensity normalized to GAPDH from two independent experiments. **d** Scheme of the iMC-linker. **e** mRNA levels of ERM proteins in wild type and shRNA ERM silenced organoids. Data are represented as the mean ± s.d. from two independent experiments. **f** LGR5+ cells treated with doxycycline to induce the iMC-linker expression, untreated (no doxy) and LGR5-ERM silenced cells on 3 kPa gel substrates. Representative images of two independent experiments. Scale bar, 50 μm. **g** Cell roundness for sorted cells seeded on 3 kPa gels. Data are represented as the mean ± s.d. of *n* ≥ 58 cells/condition from two independent experiments. Statistical significance was determined using Shapiro-Wilk normality test, followed by a Kruskal−Wallis multiple-comparison test. **h** Representative images of LGR5-, LGR5+ and iMC-linker expressing LGR5+ cells under confinement. Scale bar, 10 μm. **i** Percentage of dying/dead, polarized blebs, rounded blebs and no blebs in LGR5-, LGR5+ and iMC-linker expressing LGR5+ cells. Data are represented as the mean ± s.d. of percentages from four independent experiments (*n* ≥ 119 cells/subgroup). Statistical significance was determined using two-way analysis of variance, followed by a Šidák multiple-comparison test. **j** Migration velocity of LGR5$^{high}$, LGR5$^{low}$, iMC-linker expressing LGR5$^{high}$ clusters and LGR5$^{low}$ clusters with ERM silencing. Data are represented as the mean ± s.d. of *n* ≥ 46 clusters/condition from two independent experiments. Statistical significance was determined using Shapiro-Wilk normality test, followed by a Kruskal−Wallis multiple-comparison test. **k** Time lapse of LGR5$^{high}$, LGR5$^{low}$, iMC-linker expressing LGR5$^{high}$ and ERM silenced LGR5$^{low}$ clusters seeded on Collagen-I coated 3 kPa gels. Representative images of two independent experiments. Scale bars, 50 μm. Source data are provided as a Source Data file.

Confinement of LGR5+ cells expressing the synthetic iMC-linker resulted in responses remarkably similar to the LGR5- cells, both in terms of amoeboid cell polarization and cell death (Fig. 5h, i, Supplementary Movie 11). These data support that higher MCA in the LGR5- cells reduces their ability to adopt an amoeboid-like behavior upon cell confinement and makes them more susceptible to cell death.

To explore whether MCA explains mechanical differences at the cluster level, we induced expression of the iMC-linker in self-assembled PDOs. Clusters expressing simultaneously high levels of both LGR5 and the iMC-linker presented a rounded non-spread morphology and moved faster compared to LGR5$^{high}$ clusters not expressing the iMC-linker (Fig. 5j, k), thus supporting the involvement of higher MCA in shaping CRC cluster phenotype. Together, these observations establish the involvement of ERMs and MCA in defining the mechanical phenotypes of LGR5+ and LGR5- cells, both at the single cell and cluster levels.

### Similar mechanical phenotypes are observed in a different PDO model with a different mutational landscape

To assess whether analogous mechanical phenotypes are present in other patient derived-models, we performed our main single cell experiments using another PDO (PDO-p18) carrying only an inactivating APC mutation and a loss of function mutation in the p53 pathway[37,74]. Despite the differences in mutational burden between the two tumor organoid models, we found similar mechanical phenotypes in terms of single cell spreading and morphology, as indicated by a lower cell roundness of the LGR5+ and LGR5$^{med}$ cells compared to the LGR5- cells (Supplementary Fig. 3a−c). We then studied the cell responses to confinement, finding that similarly to PDO7, PDO-p18 LGR5+ cells also showed higher propensity to bleb under confinement while LGR5- cells were more vulnerable to confinement-induced cell death (Supplementary Fig. 3d, e). Furthermore, we observed an analogous migratory behavior of LGR5- polarized cells, displaying higher motility (Supplementary Fig. 3f). Finally, we found upregulation of two out of the three ERM proteins (Ezrin and Radixin) in the PDO-p18 LGR5- cells (Supplementary Fig. 3g). The overall similarity in mechanical phenotype between the two PDOs supports the generality of our findings.

### Expression of LGR5 and Ezrin is anticorrelated in transcriptomic data from CRC patients

To study whether the relationship between LGR5 and ERM proteins expression found in our PDO model reflects a general trait of CRC tumors, we analyzed single-cell transcriptomic data from the Samsung Medical Center (SMC) cohort[75]. For each patient dataset, we divided the cells into two groups based on LGR5 expression (Fig. 6a) and the mean expression of the ERM proteins was calculated for each group.

This analysis revealed a significant upregulation of Ezrin in the LGR5- cells (Fig. 6b, c). Although the degree of upregulation showed considerable variability between patients, it was consistent across almost all patients (Fig. 6c). To further understand the link between Ezrin and cancer differentiation, we analyzed its expression as a function of *Lgr5* and *Krt20*. This analysis showed a consistent upregulation in the KRT20+ cells, supporting that Ezrin expression is linked to cell differentiation in CRC (Supplementary Fig. 5a). Interestingly, we only found upregulation of Moesin and Radixin in the LGR5- cells in a subset of the patients (Supplementary Figs. 4a−e, 5b, c), suggesting that the negative correlation between these two proteins and cancer stemness is not as widespread as in the case of Ezrin.

We finally investigated whether the differences in ERMs expression and activity found in PDOs and in the patient cohort are specific to CRC or are a general feature of biology of the healthy tissue. Analysis of sc-RNAseq data from healthy samples revealed that Ezrin expression was found to be widespread in most of the normal colon epithelial cells, especially in colonocytes and Bestrophin 4 (BEST4 + ) cells (Fig. 6e). Radixin was found to be mostly expressed by enteroendocrine cells (EECs) (Fig. 6f) while Moesin had an overall low expression and was mostly expressed by Tuft cells and BEST2+ Goblet cells (Fig. 6g). Together, these observations support our conclusion that loss of cancer stemness in colorectal cancer is coupled with an upregulation of the ERM proteins, reflecting a molecular program conserved from the normal colon epithelium.

## Discussion

In the present study we establish that expression levels of LGR5 mark distinct mechanical phenotypes in patient-derived colorectal cancer organoids (Fig. 6h). LGR5+ cells are stiffer, adhere better to the ECM, move slower both as single cells and clusters, display nuclear YAP, and show a high survival rate in response to mechanical confinement. These traits together define a phenotype of mechanical stability and resilience. Conversely, LGR5- cells are softer, less adhesive, and faster, thus displaying a mechanical phenotype corresponding to a more dynamic state. These distinct mechanical features may favor different functions for LGR5+ and LGR5- in the metastatic cascade. The faster, softer, and less adhesive LGR5- cells likely have an advantage to escape the primary tumor and squeeze through the stroma to reach the vasculature. This can explain observations in mice showing that the majority of disseminating cells are LGR5- and that genetic ablation of a subpopulation of LGR5- cells prevents metastatic disease[11,14]. By contrast, the higher adhesion, stiffness, resilience, and nuclear YAP displayed by LGR5+ cells are all suitable mechanobiological features to potentially promote growth at a metastatic site, consistent with studies showing that long term metastatic growth is provided by LGR5+ cells[3–5,10]. We also observed that clusters of LGR5+ cells display higher

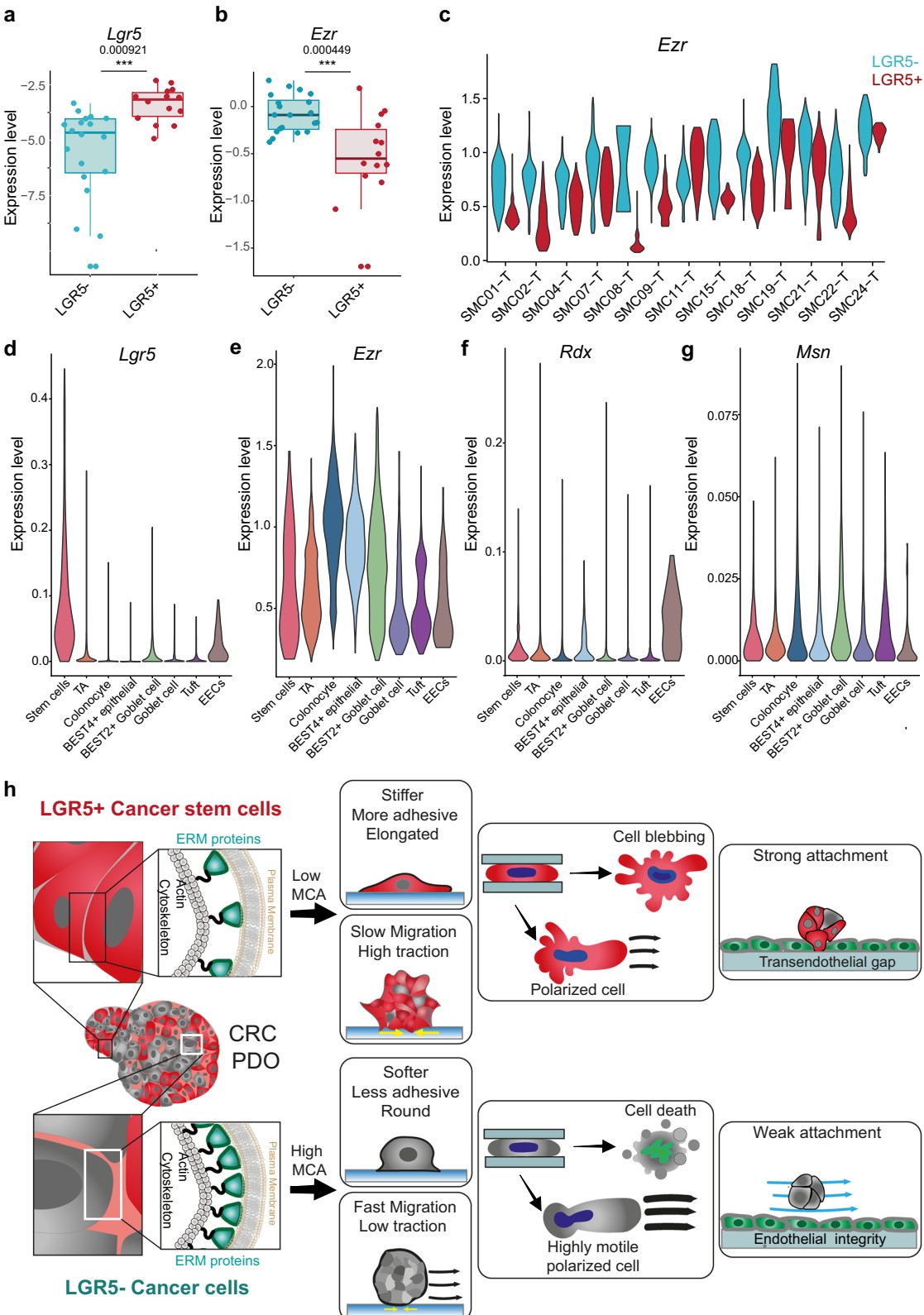

**Fig. 6 | Expression of LGR5 and Ezrin are anticorrelated in transcriptomic data from CRC patients. a**, **b** Gene expression of *Lgr5* and *Ezr* in epithelial tumor cells from CRC patients in the SMC cohort. Each dot corresponds to the average expression levels of one patient. The box center line represents the median, and the limits represent the first and third quartiles. Whiskers indicate maximum and minimum values. *n* = 20. A linear model was fitted to the data to assess significance. **c** Violin plots showing *Ezr* expression levels in single epithelial tumor cells from patients in the SMC cohort. **d**–**g** Expression of *Lgr5, Ezr, Rdx* and *Msn* in colon epithelial cells from healthy donors. **h** Summary scheme showing how membrane to cortex attachment determines different mechanical phenotypes in LGR5+ cancer stem cells and LGR5- cancer cells.

adhesion to the endothelial surface and form transendothelial gaps with higher efficiency than LGR5- cells, suggesting that LGR5+ cells have an increase ability to extravasate. This feature of LGR5+ cells may explain that when LGR5+ and LGR5- are injected in equal amounts into mice, LGR5+ cells are more efficient at seeding metastases[11]. Together with previous in vivo studies, our in vitro results indicate that differences in mechanical phenotype may provide LGR5+ and LGR5- with differential roles in metastasis; the mechanical features of LGR5- cells are suitable for dissemination from the primary tumor, whereas those of LGR5+ are suitable for extravasation and growth at secondary sites.

We show that these differences in mechanical phenotype can be explained by the upregulation of ERM proteins in LGR5- cells. ERMs are highly conserved, homologous proteins that function as linkers between the plasma membrane and the actin cortex. During homeostasis of the healthy intestinal epithelium, Ezrin expression increases as intestinal stem cells differentiate into enterocytes or colonocytes, promoting epithelial polarity and the formation of microvilli[76–78]. Our analysis of transcriptomic data from a patient cohort and PDOs showed that the inverse correlation between LGR5 and ERMs expression characteristic of development and homeostasis is retained in tumors. To study whether ERMs expression could explain the observed differences in mechanical phenotypes, we expressed a synthetic linker in LGR5+ cells that mimics the ERM tethering activity in LGR5- cells. Upon doing so, LGR5+ and LGR5- became mechanically indistinguishable, indicating that the ERMs are mainly responsible for the mechanical differences between LGR5+ and LGR5- cells. Conversely, mechanical differences were also reduced when ERM proteins were downregulated in LGR5- cells. From a mechanistic perspective, ERMs have been shown to impair both blebbing and protrusion-based spreading by preventing membrane to cortex separation[64,67,70,71,79]. Moreover, membrane-cortex tethering has recently been shown to soften the cell cortex by downregulating formin activity[80]. How ERMs may promote cell death in response to mechanical confinement is less explored, but we speculate that higher tethering increases membrane tension and favors its fracture in response to large deformations. Our study uncovers a link between colorectal cancer stemness and cell surface mechanics.

Cancer cell invasion has been attributed to the migration of both individual cells and cell groups[11,14,53,81–83]. Here we found that single cells and multicellular clusters share some common mechanical traits dependent on their expression of LGR5. Both single cells and clusters expressing low levels of LGR5 are less adherent to collagen-I substrates and adopt a rounder morphology. However, clusters with different levels of LGR5 expression showed mechanical differences that were absent at the single cell level. Unlike single cells, clusters with lower LGR5 content exerted lower forces on the substrate and moved faster, two features that may provide them with an advantage in invasion. From a physical perspective, the differences in mechanical phenotypes can be interpreted in terms of the theory of active wetting, which was recently shown to explain a coupling between cluster spreading and migration[61]. Owing to their lower surface tension, which likely originates from lower MCA, LGR5+ cells and clusters are able to wet the surfaces such as collagen-I coated substrates or endothelial monolayers. By contrast, LGR5- cells and clusters are close to a neutral wetting regime, which favors their migration and may render them more sensitive to microenvironmental gradients[63].

Along the metastatic cascade, cancer cells encounter mechanically heterogeneous microenvironments[84,85]. To survive and successfully metastasize, they need to fine-tune their mechanical properties, adapting to the dynamic physical forces they are subjected to. Each one of the mechanical states uncovered by our data, the stable and resilient LGR5+ state in contrast to the dynamic and fragile LGR5- state, may offer an advantage in certain steps of the metastatic cascade but may be detrimental in others. As cancer differentiation is plastic and dependent on microenvironmental stimuli, a transition between mechanical states may provide an adaptive mechanism to cope with changing microenvironments throughout the metastatic journey. Hence, we propose that mechanical adaptability coupled with cancer cell plasticity may be a crucial mechanism for metastatic progression.

## Methods

### Patient derived organoids culture
The research performed in this study complies with all relevant ethical regulations. The use of patient-derived colorectal cancer organoids was approved by the Ethical Committee from Hospital Clinic in Barcelona, which acts as the ethical committee for IRB Barcelona. The PDO model used in most of the in vitro experiments has been previously described and referred to as PDO7 in ref. 37. Briefly, the PDO model was engineered by knock-in CRISPR/Cas9- mediated homologous recombination. The IRES-iCasp9-T2A-TdTomato-WPRE-BGHpolyA construct was inserted after the stop codon of the *Lgr5* gene to fluorescently label LGR5+ cells using the LGR5 endogenous promoter as a driver of TdTomato expression. The PDO model carries genetic alterations in four main pathways driving colorectal carcinogenesis, namely WNT pathway (Adenomatous Polyposis Coli (*APC*) loss of function mutation), epidermal growth factor receptor (EGFR) signaling (activating KRAS G13D mutation), transforming growth factor beta (TGF-β) signaling (SMAD4 loss of function) and p53 tumor suppression (mutation in ATM). The PDO in Supplementary Fig. 3, PDO-p18, carries inactivating APC mutation and functional inactivation of TP53 mutation[74]. Tumor cells were grown as organoids embedded in basement membrane extract (Cultrex BME Type 2, AMSbio) using tumor organoid medium composed of Advanced DMEM/F12 (Gibco), 10 mM HEPES (Sigma-Aldrich), 1% GlutaMax; 1× B-27, 20 ng ml⁻¹ Human FGF (fibroblast growth factor) basic (all from Gibco); 50 ng ml⁻¹ recombinant human EGF (epidermal growth factor) and recombinant Noggin (100 ng ml⁻¹) (both from Peprotech). The medium was supplemented with 0.2% Normocin (InvivoGen) as an antimicrobial agent. The organoids were split every 6–7 d. For splitting, the organoid-containing drops were enzymatically dissociated by TrypLE (Gibco) for 15 min at 37 °C and reduced to single cell suspension by pipetting. TrypLE was then diluted with washing medium (Advanced DMEM/F12, 10 mM HEPES (Sigma-Aldrich), 1% GlutaMax) and centrifuged at 100 g at RT for 3.5 min. The pellet was resuspended in 1:3 Medium:BME and seeded in 20 µl drops in 6-wells plates. After incubation for 20 min at 37 °C, the drops were covered with tumor organoid medium and maintained at 37 °C and 5% $CO_2$.

### Flow cytometry analysis and sorting
For single cell experiments, PDOs were dissociated as described above and resuspended with cold tumor organoid medium at a concentration of $1 \times 10^6$ cells ml⁻¹. Single cells suspension was stained with DAPI (Life technologies) for 10 min, then analyzed and sorted with FACSAriaFusion flow cytometer (Beckton Dickinson) (Fig. 1b). The gating strategy defining the LGR5 + , LGR5med and LGR5- subpopulation is shown in Supplementary Fig. 1b and Supplementary Fig. 6. Briefly, cells were selected according to the FSC/SSC parameters. Aggregates were discarded using FSC-width while dead cells were excluded based on DAPI staining. Gating to select the LGR5- cells was set using unlabeled PDO cells. LGR5+ and LGR5med were selected to prevent overlapping between the two populations and ensure collection of approximately the same number of cells. After sorting, cells were centrifuged, resuspended in 50 µl tumor organoid medium and seeded on gel substrates coated with collagen I. After 3 h incubation to allow the cells to adhere, 1 ml of medium was added in each dish/ well. Maintenance of cell identity after dissociation was assessed by measuring LGR5 fluorescence of sorted single cells 24 h after sorting and seeding

(Supplementary Fig. 1c). All single cell experiments were performed 24 h after seeding and sorting.

## Polyacrylamide gel preparation

Polyacrylamide gels (PAA) were used to form substrates with different Young's modulus ranging from 0.5 to 30 kPa. Glass-bottom 35 mm dishes or 6-well glass-bottom plates (Mattek) were incubated with a solution of Bind-silane (Sigma-Aldrich), acetic acid (Sigma-Aldrich) and absolute ethanol (PanReac) at volume proportions of 1:1:12 for 10 min at RT. After 2 washes with absolute ethanol, 20 µl of polyacrylamide solution (Supplementary Table 3) were placed on the dish glass bottom and covered with 18 mm glass coverslip. For TFM, the gel substrates contained 0.2 µm green, fluorescent carboxylate-modified beads (FluoSpheres, Thermofisher). After 1 h polymerization at RT, the gels were covered with PBS and the coverslips removed. The gel surface was activated with Sulfo-SANPAH and coated with 150 µg.ml$^{-1}$ of Collagen I overnight.

## Mechanical characterization of cells using RT-DC

Real-time deformability cytometry measurements of PDO single cells was performed as previously described[45]. Briefly, cells grown for 6−7 days in BME or seeded as single cells on PAA gel substrates were harvested using TrypLE and centrifuged at 400 g for 4 min. The pellet was then suspended in a viscous solution containing 0.5% methylcellulose with osmolarity pf 310−315 mOsm kg$^{-1}$ and loaded in the RT-DC microfluidic chip using a syringe pump (NemeSys, Cetoni). Cells were flowed through a 300 µm long channel with a square cross-section of $20 \times 20$ µm at a speed of 0.16 µl s$^{-1}$. Deformed cells were imaged at the end of the channel and their cell contours used to calculate the cell deformation by ShapeIn2 software. Calculation of the apparent elastic modulus was performed using the analysis software Shape-Out version 2.11.5 (available at https://github.com/ZELLMECHANIK-DRESDEN/ShapeOut2)[86]. Cells with porosity higher than 1.05 or with an area outside the range of 150−350 µm$^2$ were discarded to avoid incomplete contours and cell clusters. Measured cells were divided into populations of LGR5-, LGR5$^{med}$ and LGR5+ based on Tdtomato fluorescence intensity as shown in Supplementary Fig. 2a.

Deformation is defined as $1 - (2*\sqrt{\pi A}/P)$, where A and P are the area and perimeter of the detected cell contour, respectively.

## Cell confinement

Cell confinement was achieved using a previously described dynamic cell confiner[52]. Briefly, sorted LGR5- and LGR5+ cells (either PDO7 or PDO-p18) at a ratio of 1:1, were confined using a microfabricated device consisting of a central polydimethylsiloxane (PDMS) piston that functions as a suction cup and is connected to a system made of a pressure controller (Flow EZ™ LU-FEZ-N800, Fluigent) and a vacuum pump (LABOPORT N96). The confinement height (4.5 and 7 µm) was controlled using custom-made microconfinement glass coverslips with PDMS micropillars of a defined height, produced as previously described[52]. All the surfaces of the device were plasma cleaned and coated with 0.5 mg ml$^{-1}$ of PLL-g-PEG to create low attachment conditions. To track confined nuclei and detect dead/dying cells, cells were incubated for 20 min with ReadyProbes Cell Viability Imaging Kit (Blue/Green).

## Cluster motility assay

PDOs grown in BME for 5 days were extracted from matrix using Dispase I. After neutralization through dilution with 10 ml washing medium, they were centrifuged and resuspended in 1 ml tumor organoid medium. A volume of 70 µl of the cluster suspension was placed on an 18 mm 3 kPa PAA gel substrates coated with 150 µg/ml of Collagen I (Fig. 3a). After 3 h incubation at 37 °C to allow clusters adhesion, 1 ml medium was added to the dish. 24 h after seeding, clusters were imaged every 60 min, for a total of 42 h.

## HUVEC monolayer formation and cluster attachment assay

To form a monolayer, HUVEC (Lonza) cells were seeded on 3 kPa PAA gel substrates coated with 150 µg/ml of Collagen I and cultured for 6 days. EGM-2 endothelial medium was changed every two days. Staining with VE-cadherin confirmed the presence of stable cell-cell junctions after 5−6 days (Supplementary Fig. 2f). Once a monolayer was formed, PDO clusters were extracted from BME matrix with Dispase I and seeded on top of the monolayer. The samples were imaged 1 h after seeding every 40 min for 16 h.

## Quantitative PCR (qPCR)

To measure the expression of ERM proteins in sorted PDO cells (PDO7 or PDO-P18), real-time qPCR experiments were performed. RNA was extracted using the Qiagen rNeasy Micro Kit and following the manufacturer's instructions (Qiagen). Concentration of the obtained total mRNA was measured with a Nanodrop ND-1000 Spectrophotometer and equal amounts were loaded for reverse transcription. Complementary DNA was produced using the iScript cDNA Synthesis Kit. SYBR Green (Applied Biosystems 4385612) RT−qPCRs were performed in triplicates or duplicates with a StepOnePlus System (Applied Biosystems) under standard settings. The 2 − ΔΔCt method was used to calculate relative gene expression. Normalization of all the ΔΔCt values was carried out to the housekeeping gene GAPDH. Supplementary Table 4 details the primer sequences used.

## Western blotting

Sorted PDO cells (-100,000) were seeded on 3 kPa PAA gel substrates. After 24 h, cells were mechanically dissociated from gel substrates using cell scrapers (Biologix) and lysed with RIPA (Radio-Immunoprecipitation Assay) Buffer (Sigma-Aldrich) containing proteases and phosphatases inhibitors (Thermo Fisher). Cell lysates were homogenized, sonicated and centrifuged at 20,000 x g for 20 min. Laemli buffer was added to cell lysates and samples were heated at 95 °C for 5 min. Proteins were separated in 4−20% polyacrylamide gels (Bio-rad) by electrophoresis. Then proteins were transferred to a nitrocellulose membrane (Whatman, GE Healthcare Life Sciences). After transfer, membranes were blocked with 5% dry milk-Tris buffer saline-0.2% Tween and incubated with primary antibodies, p-Ezrin/ Radixin/ Moesin Rabbit (cat. no. 3726, Cell Signaling Technology), total Ezrin/ Radixin/ Moesin Rabbit (cat. no. 3142, Cell Signaling Technology) and GAPDH antibody (cat. no. 5174 S, Cell Signaling) overnight at 4 °C. Membranes were incubated with horseradish-peroxidase-coupled secondary antibodies for 2 h at RT. Bands were revealed using Limi-Light kit (Roche), visualized with ImageQuant LAS 4000 and quantified using ImageJ/Fiji software.

## Bulk RNA-seq and analysis

We used RNA-seq to characterize the transcriptomic profile of sorted PDO single cells. Organoids grown in BME for 6−7 days were harvested using TrypLE and centrifuged at 400 g for 4 min. Cells were then resuspended in cold tumor organoid medium and sorted into LGR5-, LGR5$^{med}$ and LGR5+ cells (two biological replicates for each condition). At least 20,000 cells were collected per condition. RNA was extracted using the Qiagen rNeasy Micro Kit and following the manufacturer's instructions (Qiagen). RNA-seq reads from datasets were aligned with STAR (v.2.5.2)[87]. SAM files were converted to BAM and sorted using Sambamba (v.0.7.1)[88]. Gene-wise differential expression analysis was performed using the R package DESeq2 (v.1.30.1)[89]. Normalized values for plots were obtained using the rlog function of the same package. Enrichment analysis was performed using ROAST-GSA methods that combines the statistical inference based on limma[90] and rotations from ROAST[91] and the re-standardized MaxMean statistic from GSA[92]. In the plots positive values (in red) indicate gene sets enriched in the population indicated first in the legend. Negative values (in blue) indicate enrichment in the population indicated second in the legend.

### Generation of stable PDO line expressing inducible iMC-linker

A PiggyBac vector with Neomycin resistance gene and an inducible iMC-linker Halo sequence (JHL9-pPB-tetON-lynlinker-Halo-CH(utr) _optm) was co-nucleofected with a PiggyBac transposase plasmid at volume proportions of 1:2. Nucleofection was performed using Lonza nucleofector kit V (VVCA-1003) and the Lonza-AmaxaII device with program A-32 following manufacturer instructions. 72 h after nucleo-fection, successfully nucleofected PDO were selected with 400 µg ml⁻¹ neomycin (G418, Invitrogen). Vector expression was induced 16 h before experiments by adding 1 µg ml⁻¹ Doxycycline. iMC-linker was labeled using HaloTag Oregon Green Ligand (Promega). Labeled cells expressing the linker were selected and sorted FACS Aria flow cyt-ometer (BD Bioscience).

### Generation of stable PDO line expressing shRNA targeting ERM proteins

The genetic constructs to generate shRNAs targeting ezrin, radixin and moesin were stably included in the PDOs via lentiviral transduction. First, the pertinent plasmids (Addgene plasmid # 8453 containing the targeting shRNA sequences listed in Supplementary Table 2) were transiently transfected via lipofection (Lipofectamine 3000, Thermo-Fisher) to Hek293T cells, together with the corresponding envelope and packaging vectors. Second, PDOs were incubated with the lenti-viral particles obtained under sustained selection with puromycin 2 µg ml⁻¹.

### Immunostainings

To determine both protein presence and localization we used protein immunostaining. PDO single cells were fixed in 4% paraformaldehyde (PFA; Electron Microscopy Sciences) for 10 min at RT and washed three times with PBS. The samples were permeabilized with 0.1% Triton X-100 (Sigma-Aldrich) for 10 min at RT. After three washes with PBS, the samples were blocked with PBS containing 1% bovine serum albumin (BSA Sigma-Aldrich) for 1 h at RT to prevent any non-specific bonding. Primary antibodies diluted in PBS containing 1% BSA were added and incubated overnight at 4 °C. After three more washes in PBS, secondary antibodies and phalloidin in PBS were added for 2 h at RT. Finally, the samples were washed five times with PBS (5 min each) and imaged. PDO clusters were fixed with 4% paraformaldehyde for 15 min and permeabilized in 0.1% Triton PBS X-100 for 30 min. The other steps were performed as explained above for single cells.

### Antibodies

The following is a list of the primary antibodies used and their respective dilutions were: mouse anti-YAP, 1:200 (Santa Cruz, cat. no. sc-271134); mouse anti-CK20, 1:100 (Dako, cat. no. M7019); rabbit anti-VE-cadherin, 1:2000 (Invitrogen, cat. no. PA5-19612), rabbit anti-pMLC P-Myosin Light Chain 2 (Thr18/Ser19), 1:200 (Cell Signaling cat. no. 3674), rabbit anti-. Ezrin/Radixin/Moesin, 1:100 (Cell Signaling cat. no. 3142), rabbit anti-phospho Ezrin (Thr567)/ Radixin (Thr564)/ Moesin (Thr558), 1:100 (Cell Signaling cat. no. 3141). The secondary antibodies used were: goat anti-mouse Alexa Fluor 488 (Thermo Fisher Scientific, cat. no. A-11029); donkey anti-rabbit Alexa Fluor 488 (Thermo Fisher Scientific, cat. no. A-21206), goat anti-rabbit Alexa Fluor 555 (Thermo Fisher Scientific, cat. no. A-21429) and goat anti-mouse Alexa Fluor 405 (Abcam, cat. no. ab175660). All the secondary antibodies were used at a dilution of 1:400. To label F-actin, phalloidin Atto 488 (Sigma-Aldrich cat. no. 49409) was used at 1:500 and phalloidin Alexa Fluor-647 (Thermo Fisher Scientific, cat. no. A22287) at 1:400. Hoechst (Thermo Fisher Scientific, cat. no. 33342) was used to label nuclei.

### Image acquisition

For single cells mean traction and morphology measurements, the experiments were performed on an automated inverted microscope (Nikon Eclipse Ti) using a 20× 0.75 NA objective. To allow higher resolution of traction force distribution, multipole analysis was per-formed using an inverted Nikon microscope with a spinning disk confocal unit (CSU-WD, Yokogawa) and a Zyla sCMOS camera (Andor, image size 2048 × 2048 pixels). The same microscope was used for multidimensional acquisitions of confined sorted PDO single cells, cluster mobility on 2D gel substrates, cluster attachment to an endo-thelial monolayer. A Nikon ×40 × 0.75 NA air lens objective was used for multipole experiments and clusters experiments. Time-lapses of confined cells were acquired using a ×60 objective (plan apo; NA, 1.2; water immersion). High resolution images of cell clusters, either with live labeling or immunostained were acquired using a ×60 objective (plan apo; NA, 1.2; water immersion) with a z step of 0.25 or 1 µm. For all the live imaging experiments, a temperature box maintaining 37 °C in the microscope (Life Imaging Services) and a chamber maintaining CO₂ and humidity (Life Imaging services) were used. The open-source Micromanager[93] was used to carry out multidimensional acquisitions with a custom-made script.

### Cell and cluster segmentation

Custom-made MATLAB scripts were used to segment nuclei, cells and clusters. For YAP nuclear to cytoplasmic ratio quantification, fluor-escent images of stained nuclei and actin were converted to binary images using level and locally adaptive thresholding. Images were further processed through morphological operations to obtain representative binary masks of detected objects. To obtain binary images of the cytoplasm only, inverted binary images of nuclei were multiplied with masks of segmented cells. For cluster segmentation, confocal phase contrast images from a plane 2 µm above the gel sub-strates plane were processed similarly. All obtained masks were inspected and incorrectly segmented cells/ clusters were discarded.

### LGR5+ cells Tdtomato fluorescence intensity measurements

Tdtomato fluorescence intensity was quantified using ImageJ/Fiji software or MATLAB scripts. All the values were subtracted with background levels. In the single cell analysis, a mask of the cell borders was either manually drawn using phase-contrast images or obtained using custom-made MATLAB scripts and used to measure the mean fluorescence of Tdtomato. In the cluster analysis, the binary mask obtained from phase-contrast images (see section cell and cluster segmentation) was used to measure the mean gray values of the focal plane located 2 µm above the gel substrate. In the analysis of cluster attachment to an endothelial monolayer, for each cluster, the mean gray value of 4 focal planes at 0, 5, 10, 15 µm from the monolayer, was calculated and reported as Tdtomato fluorescence intensity.

### YAP nuclear to cytoplasmic ratio quantification

Segmented images of nuclei and cytoplasm were used to measure the mean fluorescence intensity of YAP. A ratio of the measured intensities was then calculated.

### Shape analysis

Single cells and cluster roundness was measured with the shape descriptor tool in Imagej/Fiji software.

### Cell and clusters tracking

Single cell velocities were obtained as previously described[94]. Single cell trajectories from 16 h time-lapses acquired every 20 min were tracked using the Manual Tracking plug-in from ImageJ/Fiji. Quantifi-cation of cell velocity calculated as the accumulated distance/total time acquired, was performed and analyzed using the Chemotaxis Tool plug-ins from ImageJ/Fiji. Confined nuclei and cluster velocities were computed using custom-made MATLAB scripts as previously described[63]. In each frame, the centroid position of segmented nuclei/ clusters was detected. Trajectories were built based on proximity and

velocity was calculated from the distance between the centroids in each frame. Inconsistent or inaccurate tracks were discarded.

## Cluster contact angle and 3D reconstruction

Cluster contact angle with the substrate was calculated as previously described[63]. Briefly, for dewet clusters the angle was calculated as a function of the contact radius ($R$, the radius of the plane in contact with the substrate) and the cluster radius ($R_{sphere}$) using the formula $\alpha = 180 - asin\frac{R}{R_{sphere}}$. For wet clusters the formula used was $\theta = asin\frac{R}{R_{sphere}}$ with $R_{sphere}$ calculated as $R_{sphere} = \frac{R^2 + H^2}{2H}$ where $H$ is the cluster height. 3D cluster rendering from high resolution z stacks of labeled actin was obtained using Imaris (version 9.1.0) software. A Gaussian filter was applied to the images to smooth the fluorescence signal before generating a surface to visualize the 3D shapes of clusters.

## Traction force microscopy

All traction force microscopy experiments were performed using cells seeded on polyacrylamide gel substrates of known stiffness, containing fluorescent beads at a concentration of 0.03 w/v. Traction computations and the following analyses of traction forces were carried out with custom-written MATLAB scripts. Fourier transform traction microscopy was used to measure traction forces[94–96]. The displacement fields of the fluorescence microspheres were obtained using a home-made particle imaging velocimetry algorithm (PIV) using square interrogation windows of side 32 pixels with an overlap of 0.5.

## Multipole analysis of tractions

The dipole moment of a cell was calculated as described by Tanimoto and Sano[44]. Cells were manually segmented from the bright field channel, and a binary mask was obtained. This mask was applied to the cell's traction field, to keep only the traction forces generated by the cell and suppress any spurious tractions. Following this, the cell's dipole moment was calculated and diagonalized, yielding eigenvalues corresponding to the major and minor dipole magnitudes. The anisotropy of the cell's mechanical response was quantified as the ratio between the major and minor dipole eigenvalues[97].

## Patient 10x single-cell analysis

Count matrices were downloaded from ArrayExpress (E-MTAB-8107 for samples EXT001, EXT002, EXT003, EXT009, EXT010, EXT011, EXT012, EXT013, EXT014, EXT018, EXT019, EXT020, EXT021, EXT022, EXT023, EXT024, EXT025, EXT026, EXT027 and EXT028) and GEO (GSE132465 for all Samsung Medical Center tumor (SMC-T) samples)[75]. The remaining EXT samples were processed as referred in E-MTAB-8107 and deposited in ArrayExpress under accession number E-MTAB-9934. Data was processed as detailed in ref. 14. Expression was imputed and smoothened using the MAGIC algorithm[98]. Signature scores were computed as the mean value of the MAGIC expression per cell for all genes in the signature. The LGR5+ cell population was defined as cells with the LGR5 signature expression (*Bcl11B, Axin2, Lgr5, Ascl2, Lrig1*) above the 75th percentile. Patient datasets where the LGR5+ cells could not be distinguished from the LGR5- population were not included by thresholding in the quantification of gene expression. Data from healthy human samples are from the gut atlas human dataset (https://www.gutcellatlas.org). All the datasets were subsetted to include only the colon tissue and the epithelial partition in the analysis.

## Statistical analysis

All the plots were generated in GraphPad Prism 9. All images and videos were processed with the open-source software ImageJ/Fiji. All data are represented as the mean ± s.d. Sample size (n) and the number of independent repetitions is indicated in the figure captions or in Supplementary Data 2. Statistical analyses were performed using GraphPad Prism 9. A normality and lognormality test were used to establish the appropriate significance test, followed by a statistical test to compare the mean. For data with more than one variable, analysis of variance tests (ANOVA) or mixed effects analysis followed by multiple comparisons tests were applied.

## Reporting summary

Further information on research design is available in the Nature Portfolio Reporting Summary linked to this article.

## Data availability

The RNA-seq data generated in this study are publicly available at the Gene Expression Omnibus (GEO) with the accession number: GSE247359. Previously published single cell RNA sequencing data of CRC patient samples were reanalyzed and are available at GEO under accession codes GSE132465[75] Count matrices for single-cell RNA-seq experiments were deposited at ArrayExpress under accession number E-MTAB-8107 and E-MTAB-9934. Data from healthy human samples[99] can be found in the gut atlas human dataset (https://www.gutcellatlas.org) with accession number E-MTAB-8901. The remaining data are available within the Article, Supplementary Information or Source Data file. The source data underlying Figs. 1c, f–h, j, 2c, d, g, i, 3c–f, h, 4c–f, 5a–c, e, g, i, j, 6a, b, and Supplementary Figs. 1a, c–h, j–l, 2a, b, d, e, g, h, 3b, c, e, f, g, 4a, b are provided as a Source Data file. Source data are provided with this paper.

## Code availability

Code generated for this study can be found at: https://github.com/xt-prc-lab/Conti_et_al_2024.git. Codes for tracking moving clusters were previously published[63] and are already available online.

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

## Acknowledgements

We thank all the members of the Trepat and Roca-Cusachs lab for support and discussion. We thank Anghara Menéndez Montes, Susana Usieto, Mònica Purciolas, the core facility for biostatistics at IRB Barcelona and the Cytometry Unit from Scientific and Technological Centers (CCiTUB), Universitat de Barcelona, for technical assistance. We thank Tina Haase for providing HUVEC cells. We thank Srivatsava Viswanadha Venkata Naga Sai for valuable suggestions. Finally, we thank Marija Matejcic, Alice Perucca, Gerardo Ceada, Amy Beedle for their feedback on the manuscript and suggestions. The work was funded by the Generalitat de Catalunya (AGAUR 2021 SGR 01425 to X.T. and P.R.-C., the CERCA Programme, and "ICREA Academia" award to P.R-C., AGAUR SGR-2021-001278 to E.B.); the Spanish Ministry for Science and Innovation MICCINN/FEDER (PID2021-128635NB-I00, MCIN/AEI/ 10.13039/ 501100011033 and "ERDF-EU A way of making Europe" to X.T., PID2019-110298GB-I00 to P.R.-C., Mineco BES-2017-079847 to S.C., PID2020-117011GB-I00 to V.R., PID2020-119917RB-100 to E.B., PID2021-125212OA-I00 to A.L., RYC2020/029736/I to A.L.); the European Research Council (883739 Epifold to X.T., 884623 residualCRC to E.B., 101097753 MechanoSynth to P.R.-C.), Fundació la Marató de TV3 (project 201903-30-31-32 to X.T. and E.B. and 201936-30-31 to P.R.-C.); European Commission (H2020-FETPROACT-01-2016-731957 to P.R.-C. and X.T.); La Caixa Foundation (LCF/PR/HR20/52400004 and ID 100010434 under the agreement LCF/PR/HR20/52400004 to P.R.-C. and X.T., LCF/BQ/ PR18/11640001 to A.L.); The European Molecular Biology Laboratory (to A.D.-M.); The EMBL Interdisciplinary Postdocs (EIPOD) fellowship under Marie Sklodowska-Curie Actions COFUND and Joachim Herz Foundation Add-on fellowship (to J.H.L); European Union's Horizon EIC-ESMEA Pathfinder program under grant agreement (101046620 to V.R.); and

The Peter and Traudl Engelhorn postdoctoral fellowship to C.K.X; European Union–– NextGenerationEU through the Italian Ministry of University and Research under PNRR–– M4C2-I1.3 Project PE_00000019""HEAL ITALIA (G.S.). IBEC, CRG and IRB are recipients of a Severo Ochoa Award of Excellence from the MINECO.

## Author contributions

S.C., A.L. and X.T. conceived the project. S.C. performed experiments and analyzed data. V.V. performed confinement experiments with S.C. and contributed technical expertize and discussion. C.K.X. performed RT-DC experiments with S.C.. C.C. generated the CRISPR/Cas9 LGR5-IRES-tdTomato engineered PDOs. J.F.A. generated the silenced cell lines and performed RT-PCR and WB with S.C.. C.S.-O.A. and E.M.G. analyzed human CRC transcriptomic datasets and performed scRNA seq analysis. J.H.L. generated the construct for inducible iMC-linker-Halo expression. L.R. developed analysis software and contributed technical expertize and discussion. J.H.L., A.C.S, C.C., P.R.-C., V.R., J.G., A.D.-M., E.B., A.L. and J.F.A. contributed technical expertize, materials and discussion. G.S. generated PDO7. S.C. and X.T. wrote the manuscript. All authors revised the completed manuscript. X.T supervised the project.

## Competing interests

The authors declare no competing interests.

## Additional information

[1]Institute for Bioengineering of Catalonia (IBEC), The Barcelona Institute for Science and Technology (BIST), Barcelona, Spain. [2]Centre for Genomic Regulation (CRG), The Barcelona Institute for Science and Technology (BIST), Barcelona, Spain. [3]Institute for Research in Biomedicine (IRB Barcelona), Barcelona Institute of Science and Technology (BIST), Barcelona, Spain. [4]Centro de Investigación Biomedica en Red de Cancer (CIBERONC), Barcelona, Spain. [5]Max Planck Institute for the Science of Light, Erlangen, Germany. [6]Cell Biology and Biophysics Unit, European Molecular Biology Laboratory (EMBL), Heidelberg, Germany. [7]Department of Surgical Oncological and Stomatological Sciences, University of Palermo, Palermo, Italy. [8]Facultat de Medicina, University of Barcelona (UB), Barcelona, Spain. [9]Universitat Pompeu Fabra (UPF), Barcelona, Spain. [10]Institució Catalana de Recerca i Estudis Avançats (ICREA), Barcelona, Spain. [11]Department of Physics, Friedrich-Alexander Universität Erlangen-Nürnberg (FAU), Erlangen, Germany. [12]Max-Planck Zentrum für Physik und Medizin, Erlangen, Germany. [13]Centro de Investigación Principe Felipe (CIPF), Valencia, Spain. [14]Centro de Investigación Biomédica en Red en Bioingeniería, Biomateriales y Nanomedicina (CIBER-BBN), Barcelona, Spain. ✉e-mail: eduard.batlle@irbbarcelona.org; alabernadie@cipf.es; xtrepat@ibecbarcelona.eu

