## [Peer Review File · Nature Communications]

Membrane to cortex attachment determines different mechanical phenotypes in LGR5+ and LGR5- colorectal cancer cellsEditorial Note: Parts of this Peer Review File have been redacted as indicated to maintain the confidentiality of unpublished data.

REVIEWER COMMENTS

Reviewer #1 (Remarks to the Author):

The authors sought to explore the differences between colorectal cancer cells with high or low “stemness” (high or low expression of the LGR5 marker), and focus on their mechanical properties based on the multiple possible connections between such properties and the different steps of metastasis.

To understand the differences between LGR5+ and LGR5- cells under adhesion conditions, the authors measure and observe:

- cell shape differences, retained over a range of mechanically-tunable hydrogels
- YAP localization, retained over a range of mechanically-tunable hydrogels
- no difference in migration speed
- no difference in net traction forces (apart from the direction of traction, which is more directional in LGR5+ cells likely owing to their spindle/asymmetric morphology)

To understand the response of cells to cell detachment and/or migration in low-attachment conditions, the authors measure and observe:

- different deformability in suspension, accompanied by higher overall stiffness also upon adhesion
- different responses to confinement, with LGR5+ cells forming more blebs, and LGR5- cells more prone to death. In a distinct subgroup of cells (forming polarized blebs), LGR5- cells migrated faster in response to confinement.

The authors then analyze the behavior of clusters, in keeping with the idea that CRC cells can also disseminate as clusters, and find:

- that clusters retain the roundness (LGR5-) vs. spreading (LGR5+) characteristics observed at the single-cell level
- that round LGR5- clusters have lower traction forces (likely due to higher cell-cell traction, and correspondingly lower cell-ECM traction)
- that round clusters move faster
- that spreading LGR5+ clusters not only adhere strongly to collagen, but also to an endothelial monolayer mimicking the extravasation step
- that spreading LGR5+ clusters adhering to endothelial cells are better at displacing endothelial cells

The authors then explore what drives these differential mechanical properties, by focusing on expression of ERM proteins that are known to play an important role in defining the structure and stiffness of the cell cortex. The authors find:

- higher expression of Moesin and to some extent of Radixin in LGR5-, in keeping with a more rounded shape
- higher phosphorylation of ERMs in LGR5-, in keeping with active ERMs
- that artificially fostering plasma-membrane to actin attachment is sufficient to induce cell rounding in LGR5+ cells (both as single cells and as clusters), as well as reduced blebbing and increased cell death.

Finally, the authors look for correlative evidence in single-cell transcriptomics of CRC patient samples. Here, high LGR5 expression correlates with low Ezrin expression, which is however not observed in PDO7 cells. Moreover, they only find a correlation with Moesin and Radixin in a subset of patients. Nonetheless, the authors come to the conclusion that loss of cancer stemness in CRC (LGR5⁻ status) is coupled with upregulation of ERMs.

The proposed model is that LGR5⁻ cells should be better at leaving the primary tumor, while LGR5⁺ cells should be better at growing a secondary tumor once arrived at the metastatic site.

This manuscript provides an extensive “mechanical” characterization of patient-derived CRC cells, by using different and recently developed read-outs. Clearly, these cells are different by many parameters, which overall make sense with the final finding of a differential expression of ERM proteins. However, the manuscript falls short in at least two ways: (i) the generality of such findings is unknown, and (ii) the functional relevance of such findings remains unaddressed.

Point 1. The authors describe cells derived from a single patient-derived organoid (PDO7), and show that by sorting cells based on LGR5 levels, they obtain different cell population with distinct properties. Two main question arise: A) are these phenotypes a general characteristic of the LGR5^{+/-} cells, independent of the patient of origin? Or is this limited to PDO7 only? The authors use YAP as a mechano-sensor, but find that these CRC cells are not mechano-sensitive. This is surprising, because other colorectal cell lines and colon primary cells were shown to retain mechano-sensitivity. So, the issue of generality is an important and relevant one to address. B) LGR5⁻ cells are able, both upon xenografting and upon metastatic dissemination, to reconstitute the entire complexity of the original population, including LGR5⁺ cells. It would be important to understand how long the “mechanical” phenotypes are maintained - this may imply that the same signal that maintains LGR5, “stemness” and tumor-seeding ability in these cells also actively maintains the other “mechanical” phenotypes here described. The possibility also exists that these traits are instead separated, so that LGR5⁻ cells transit to LGR5⁺ cells but maintain their “mechanical” phenotype unaltered. This, together with point 1, would also clarify the general nature of such “mechanical” heterogeneity.

Point 2. The whole point in studying potential differences between LGR5⁺ and LGR5⁻ CRC cells is that LGR5 positively correlates with stem cell traits and with tumor-seeding ability, but anti-correlates with metastatic seeding ability. The authors clearly introduce this idea, discuss their data in light of this, but never provide solid data linking these phenotypes to a cancer-relevant cell behavior (i.e. survival in the bloodstream, extravasation ability, metastasis seeding ability etc.).

The fact that Ezrin overall correlates in patients with LGR5 is interesting, but Ezrin is not different in the in vitro system used here. Conversely, in vitro Moesin is clearly different between LGR5^{+/-} cells, but this is not observed in patients. So, the in vivo situation appears different, or more complicated. Moreover, the only attempt at demonstrating some causative link between LGR5 expression, ERM expression and the “mechanical” phenotypes is based on an extremely artificial system. At a minimum, it would be important to show that interfering with endogenous ERM proteins results in a coherent regulation of the “mechanical” phenotype in the opposite way in LGR5⁻ cells vitro. This could also be used to study the specific requirement of single ERM proteins (by reconstituting only one of the members in cells lacking all of them).

Then, a prediction of the data is that LGR5⁻ cells should be less prone to adhere to an

endothelium, to traverse it, and to survive the compression associated with dissemination/extravasation compared to LGR5+ cells, depending on Moesin expression. This is however in contrast with the experimental observation that LGR5- cells are better able to seed metastasis, when these parameters are important. Conversely, the fact that LGR5+ cells are better at traversing endothelia can hardly be correlated with the observation that LGR5+ cells are better at growing a metastasis AFTER they disseminate (as LGR5- cells) into the secondary organ. So, the proposed model lacks of any real functional validation with cancer assays where the relevance of LGR5 has been established, and the importance of these observations for CRC remains unknown.

The authors should decide whether they want to expand on point 1 - by providing a robust and general picture of the mechanical properties of several patient-derived LGR5+ and LGR5- populations - or of point 2 - by providing experimental support that such mechanical properties are truly relevant, at least in these PDO7 cells, to meaningfully influence metastasis dynamics.

Reviewer #2 (Remarks to the Author):

In this paper Conti et al. studied the mechanical properties of the different pool of cancer stem cells in colorectal cancer tumors by using colorectal cancer patient derived organoids to understand if and how these differences contribute to metastatic growth. They found that LGR5+ and LGR5- cells display diverse mechanical properties with LGR5+ cells being stiffer, better adherent to ECM and slower compared to the LGR5- population both when isolated as single cells or in clusters. They also linked these differences to the downregulation of the ERM proteins, a characteristic that seems conserved throughout colorectal cancer tumors.

The study is interesting and tended to approach a recurrent problem in CRC which is the metastatic relapse even after removal of the primary tumor. How this occurs from LGR5- cells and which mechanical properties are functionally relevant for the metastatic cascade is still not well known. However, the study remains superficial regarding some aspects and points out mainly a correlation of LGR5 expression and mechanical properties while the authors did not untangle the relationship between the mechanical and biochemical properties to understand if mechanical adaptability is the main driver of the metastatic progression.

Within the study three main areas of concerns are noticed:

Major points:

(1) The authors used a previously published CRC patient derived organoid line and sorted the cells in LGR5-, LGR5med and LGR5+ cells based on the LGR5 intensities. However, the study lacks a detailed analysis of the cell type identity and cell heterogeneity of the different populations. For example, the authors cite the paper Ganesh et al., where they show that the L1CAM gene is required for metastatic colonization and its expression overlap with LGR5+ cells. However, the authors didn't do any deep characterization of the three population they used in the study. This becomes especially evident for the LGR5- population that can be divided in two distinct group even before sorting. It is not clear whether this population comprises different cell states or even different cell types. Indeed, when looking at the cluster experiments after 15h a LGR5+ cell appear. Is this a rare event and how does this influence the mechanical properties of the cluster?

(2) Regarding the migration assay: it is known that migration is a collective property and

indeed clusters move faster than single cells. Yet, it remains unclear how cells in the cluster move. A more detailed study of how cells move individually under confinement and how they move as a cluster would benefit the study. Within a migrating cluster, do all the cells show the same polarity? Do they move all together in the same direction? What mode of migration do they follow? The authors could further analyze the actin cytoskeleton and myosin and see if this can be the main difference between positive and negative clusters. Can migration be inhibited or equalized between the two populations using inhibitors of the iMC linker?

(3) The authors identify membrane to cortex attachment via ERM as important mediator of the different mechanical properties observed in LGR5+ and LGR5- cells. This is one of the key findings of this study and a better characterization of the mechanism by which ERM protein induces a change in mechanical phenotype would be important also in respect to the in vivo data. It is not clear for example if the mechanism is mediated by the difference in the phosphorylation status of ERM and their activation and one way to check this would be using chemical or genetic perturbations. This can be also used to understand if it is possible also to convert the mechanical phenotype of the LGR5- cells towards the LGR5+ cells. Moreover, the authors should check the total level of ERM together with their phosphorylation status. As the authors define ERM as the main driver of the mechanical properties of LGR5- and + populations, leading to dissemination or extravasation of the tumor, it would be important to check if the LGR5- phenotype, induced by the expression of iMC in LGR5+ cells, have further consequences like changes in cell identity, proliferation and/or migration.

(4) Very interesting is also the ERM transcriptional difference the authors find in the patient cohort. However, to strengthen the relevance of the finding a more extensive analysis of the transcriptomic data would be important. How is the expression of ERM in healthy tissue and is the difference between ERM expression a cancer specific trait that is acquired or exists also in healthy cells? Do other proteins interacting with ERM show a similar expression pattern? Why differently to the PDOs, only Ezrin upregulated in the negative LGR5 population? Within the LGR5- populations is ERM differently expressed in different cell types or differently expressed in cells that are more or less differentiated? How different is the expression of L1CAM?

Minor Points:

- a. In CRC as well as in PDOs, CK20 and LGR5 are expected to have complementary expression domains. Yet, Fig1a shows co-staining of CK20 and Lgr5. Can the authors comment on this?
- b. Can the authors comment on why the stiffnesses of the PAA gels were not kept equal throughout the entire study? In some experiments the stiffness was 0.3, 3.5 and 11 kPa and in other 0.3, 1.5, 5 and 11kPa. What is the reasoning for the choice of this range of stiffness and how relevant is this stiffness in the vivo context? Moreover, the clusters analyses were all done on a substrate of 1kPa, and the authors never show if there is any difference in shape of migration at difference stiffness. Since they show emergent mechanical properties only in multicellular clusters, maybe also the response to stiffness is different compared to single cells.
- c. As single cells LGR5- and Lgr5+ cells differ in nuclear YAP. Is this difference also present in the 3D organoids? Are differences only on the localization of YAP or also its expression level and what functional implications does this have?
- d. From FigureS2b it is not clear if in the confinement experiment the authors seeded a mixed population in the cell confiner. It would be interesting to know the % of total population at different confinement for the three different population.
- e. Shape analysis measurement on the cell clusters were done on the entire cluster and not

for each cell inside the cluster. It would be interesting to analyze cell shape in their initial 3D environment (in the PDO2) to check if the shape features are preserved after dispase treatment or sorting.

f. Regarding the style of the figures, I would suggest that the authors combine Fig5,6 and 7 in two figures.

REVIEWER COMMENTS

We thank the reviewers for their thorough analysis of our manuscript and for their valuable feedback. In response to their comments and suggestions, we have performed additional experiments and analysis, used a new PDO model and deepened our molecular characterization of the LGR5-, LGR5^{med} and LGR5+ populations both *in vitro* and at the level of patient and healthy tissue. The newly added data, plots and text improved the original manuscript and further support our conclusions.

Reviewer #1 (Remarks to the Author):

The authors sought to explore the differences between colorectal cancer cells with high or low “stemness” (high or low expression of the LGR5 marker), and focus on their mechanical properties based on the multiple possible connections between such properties and the different steps of metastasis.

To understand the differences between LGR5+ and LGR5- cells under adhesion conditions, the authors measure and observe: cell shape differences, retained over a range of mechanically-tunable hydrogels YAP localization, retained over a range of mechanically-tunable hydrogels no difference in migration speed no difference in net traction forces (apart from the direction of traction, which is more directional in LGR5+ cells likely owing to their spindle/asymmetric morphology). To understand the response of cells to cell detachment and/or migration in low-attachment conditions, the authors measure and observe: different deformability in suspension, accompanied by higher overall stiffness also upon adhesion different responses to confinement, with LGR5+ cells forming more blebs, and LGR5- cells more prone to death. In a distinct subgroup of cells (forming polarized blebs), LGR5- cells migrated faster in response to confinement.

The authors then analyze the behavior of clusters, in keeping with the idea that CRC cells can also disseminate as clusters, and find: - that clusters retain the roundness (LGR5-) vs. spreading (LGR5+) characteristics observed at the single-cell level- that round LGR5- clusters have lower traction forces (likely due to higher cell-cell traction, and correspondingly lower cell-ECM traction) that round clusters move faster that spreading LGR5+ clusters not only adhere strongly to collagen, but also to an endothelial monolayer mimicking the extravasation step that spreading LGR5+ clusters adhering to endothelial cells are better at displacing endothelial cells. The authors then explore what drives these differential mechanical properties, by focusing on expression of ERM proteins that are known to play an important role in defining the structure and stiffness of the cell cortex. The authors find: higher expression of Moesin and to some extent of Radixin in LGR5-, in keeping with a more rounded shape higher phosphorylation of ERMs in LGR5-, in keeping with active ERMs that artificially fostering plasma-membrane to actin attachment is sufficient to induce cell rounding in LGR5+ cells (both as single cells and as clusters), as well as reduced blebbing and increased cell death.

Finally, the authors look for correlative evidence in single-cell transcriptomics of CRC patient samples. Here, high LGR5 expression correlates with low Ezrin expression, which is however not observed in PDO7 cells. Moreover, they only find a correlation with Moesin and Radixin in a subset of patients. Nonetheless, the authors come to the conclusion that loss of cancer stemness in CRC (LGR5- status) is coupled with upregulation of ERMs. The proposed model is that LGR5- cells should be better at leaving the primary tumor, while LGR5+ cells should be better at growing a secondary tumor once arrived at the metastatic site.

This manuscript provides an extensive “mechanical” characterization of patient-derived CRC cells, by using different and recently developed read-outs. Clearly, these cells are different by many parameters, which overall make sense with the final finding of a differential expression of ERM proteins. However, the manuscript falls short in at least two ways: (i) the generality of such findings is unknown, and (ii) the functional relevance of such findings remains unaddressed.

Point 1. The authors describe cells derived from a single patient-derived organoid (PDO7), and show that by sorting cells based on LGR5 levels, they obtain different cell population with distinct properties. Two main question arise: A) are these phenotypes a general characteristic of the LGR5+/- cells, independent of the patient of origin? Or is this limited to PDO7 only? The authors use YAP as a mechano-sensor, but find that these CRC cells are not mechano-sensitive. This is surprising, because other colorectal cell lines and colon primary cells were shown to retain mechano-sensitivity. So, the issue of generality is an important and relevant one to address. B) LGR5- cells are able, both upon xenografting and upon metastatic dissemination, to reconstitute the entire complexity of the original population, including LGR5+ cells. It would be important to understand how long the “mechanical” phenotypes are maintained - this may imply that the same signal that maintains LGR5, “stemness” and tumor-seeding ability in these cells also actively maintains the other “mechanical” phenotypes here described. The possibility also exists that these traits are instead separated, so that LGR5- cells transit to LGR5+ cells but maintain their “mechanical” phenotype unaltered. This, together with point 1, would also clarify the general nature of such “mechanical” heterogeneity.

Point 2. The whole point in studying potential differences between LGR5+ and LGR5- CRC cells is that LGR5 positively correlates with stem cell traits and with tumor-seeding ability, but anti-correlates with metastatic seeding ability. The authors clearly introduce this idea, discuss their data in light of this, but never provide solid data linking these phenotypes to a cancer-relevant cell behavior (i.e. survival in the bloodstream, extravasation ability, metastasis seeding ability etc.). The fact that Ezrin overall correlates in patients with LGR5 is interesting, but Ezrin is not different in the in vitro system used here. Conversely, in vitro Moesin is clearly different between LGR5+/- cells, but this is not observed in patients. So, the in vivo situation appears different, or more complicated. Moreover, the only attempt at demonstrating some causative link between LGR5 expression, ERM expression and the “mechanical” phenotypes is based on an extremely artificial system. At a minimum, it would be important to show that interfering with endogenous ERM proteins results in a coherent regulation of the “mechanical” phenotype in the opposite way in LGR5- cells vitro. This could also be used to study the specific requirement of single ERM proteins (by reconstituting only one of the members in cells lacking all of them). Silencing ERM in the PDOs and assessing their levels. Then, a prediction of the data is that LGR5- cells should be less prone to adhere to an endothelium, to traverse it, and to survive the compression associated with dissemination/extravasation compared to LGR5+ cells, depending on Moesin expression. This is however in contrast with the experimental observation that LGR5- cells are better able to seed metastasis, when these parameters are important. Conversely, the fact that LGR5+ cells are better at traversing endothelia can hardly be correlated with the observation that LGR5+ cells are better at growing a metastasis AFTER they disseminate (as LGR5- cells) into the secondary organ. So, the proposed model lacks of any real functional validation with cancer assays where the relevance of LGR5 has been established, and the importance of these observations for CRC remains unknown.

The authors should decide whether they want to expand on point 1 – by providing a robust and general picture of the mechanical properties of several patient-derived LGR5+ and LGR5- populations - or of point 2 - by providing experimental support that such mechanical properties are truly relevant, at least in these PDO7 cells, to meaningfully influence metastasis dynamics.

We thank the referee for their detailed analysis of our work and for their constructive criticism. The referee raises two main points that concern generality (point 1) and mechanism (point 2). They suggest addressing one of them to strengthen our conclusions. We believe both points are relevant and decided to address each of them as detailed below.

Point 1. Generality

The referee first asks whether our findings are specific to the PDO used in our study (PDO7 quadruple mutant) or a more general result (question A). To address this question, we repeated the main single cell experiments using another PDO (PDO-p18) carrying only an inactivating APC mutation and a mutation in p53 pathway. Despite the differences in mutational burden between the two tumor organoid models, we found similar mechanical phenotypes in terms of single cell spreading and morphology, response to confinement, migratory behavior of polarized cells and upregulation of two out of the three ERM proteins (ezrin and radixin). These results support the generality of our findings. They are shown in Supplementary Fig. 3 and presented on page 10, lines 314-329.

Regarding the referee's point on mechanosensitivity, we now see our original statements were unclear. Both LGR5+ and LGR5- cells are mechanosensitive as seen, for example, by the increase in traction force generation with substrate stiffness (Fig. 1h). We and others have used this readout as a univocal signature for mechanosensitivity (Elosegui-Artola et al., 2016). Our claim in the previous manuscript about lack of mechanosensitivity was aimed at emphasizing that substrate stiffness did not change the YAP nuclear to cytoplasmic ratio (Fig 1g). However, this result does not imply lack of mechanosensitivity, and we have removed that statement. We thank the referee for raising this point.

As for the question on the stability of the mechanical phenotypes and their link to LGR5 expression (question B), we followed the changes in cell roundness and LGR5 fluorescence over time for two cell populations that started from sorted LGR5- and LGR5+ single cells. Acquisition begun 24 h after sorting and our readout was cell roundness on 3kPa gel substrates coated with Collagen I. These experiments were carried out using full tumor organoid medium (the same that was used in all the experiments) which contains factors (noggin, FGF) that promote the stem cell phenotype. Tdtomato fluorescence increased in the two populations over time and this increase was paralleled with acquisition of a more elongated phenotype, thus indicating that the transition from LGR5- to LGR5+ is coupled with the emergence of a mechanical phenotype characterized by higher adhesiveness and spreading. We conclude from this experiment that expression of LGR5 is indeed coupled with a mechanical phenotype characterized by an elongated shape. Moreover, these observations confirm previous findings indicating that the two populations are plastic and that the LGR5 phenotype is dependent to a certain extent on the niche signals, as indicated by the acquisition of LGR5+ phenotype by the LGR5- cells when cultured with stemness promoting medium. These new results are shown in Supplementary Fig. 1f-h and described in the text on page 5, lines 143-147.

Point 2. ERM mechanism

The referee points out that in contrast to the patient cohort data, where Ezrin is significantly upregulated in the LGR5- cells, the qPCR results of our *in vitro* model (PDO7) show only an upregulation of Radixin and Moesin. In our original manuscript, we indeed reported a tendency of Ezrin overexpression in LGR5- cells but without reaching the threshold of statistical significance. To clarify this result, we performed additional qPCR experiments which now establish significant Ezrin overexpression in LGR5- on solid statistical grounds. Overexpression was further confirmed by new bulk RNAseq data. Thus, both in our PDO and patient data, Ezrin is overexpressed in the LGR5- population. The updated plots are shown in Fig. 5b and described on page 9 lines 272-275.

We also followed the reviewer's suggestion of carrying out loss-of-function experiments for ERM proteins. This type of experiments is well known to be highly challenging for ERM proteins owing to their redundancy and complex regulation. Nonetheless, we attempted to create a stable PDO line expressing shRNAs that target the expression of Ezrin, Radixin and Moesin. We first performed a screening of the most efficient shRNA sequences from a list of 5 sequences for each protein.

Target Prot.	Clon eID	Target Sequence	Oligo Sequence
EZR	58	CCTGGAAATGTATGGAA TCAA	CCGGCCTGGAAATGTATGGAATCAACTCGAGTTGATTCCATAC ATTCCAGGTTTTTG
EZR	59	CCAGCCAAATACAAC TGAAG	CCGGCCAGCCAAATACAACGGAAACTCGAGTTTCCAGTTGT ATTTGGCTGGTTTTTG
EZR	60	CCCACGTCTGAGAATC AACAA	CCGGCCCACGTCTGAGAATCAACAACCGAGTTGTTGATTCT CAGACGTGGTTTTTG
EZR	61	CGTGGGATGCTCAAAG ATAAT	CCGGCGTGGGATGCTCAAAGATAATCTCGAGATTATCTTTGAG CATCCCACGTTTTTG
EZR	62	CTCCACTATGTGGATAA TAAA	CCGGCTCCACTATGTGGATAATAAACTCGAGTTTATTATCCACA TAGTGGAGTTTTTG
RDX	33	GCCTTATGTATGGGAAA CCAT	CCGGGCCTTATGTATGGGAAACCATCTCGAGATGGTTTCCCAT ACATAAGGCTTTTTG
RDX	34	CGTGTATTGGAACAACA CAAA	CCGGCGTGTATTGGAACAACACAAACTCGAGTTTGTGTTGTTT CAATACACGTTTTTG
RDX	35	GCCAGAGATGAAACCA AGAAA	CCGGGCCAGAGATGAAACCAAGAAACTCGAGTTTCTTGTTTT CATCTCTGGCTTTTTG
RDX	36	GCAGACAATTAAGCTC AGAA	CCGGGCAGACAATTAAGCTCAGAACTCGAGTTCTGAGCTTTA ATTGTCTGCTTTTTG
RDX	37	GCTAAATTCTTTCCTGA AGAT	CCGGGCTAAATTCTTTCCTGAAGATCTCGAGATCTTCAGGAAA GAATTTAGCTTTTTG
MSN	8	GCTAAATTGAAACCTGG AATT	CCGGGCTAAATTGAAACCTGGAATTCTCGAGAATTCCAGGTTT CAATTTAGCTTTTTG
MSN	9	CCAGTCTAAGTATGGCG ACTT	CCGGCCAGTCTAAGTATGGCGACTTCTCGAGAAGTCGCCATA CTTAGACTGGTTTTTG
MSN	10	GCTCTTTAAGTTCCGTG CCAA	CCGGGCTCTTTAAGTTCCGTGCCAACTCGAGTTGGCACGGAA CTTAAAGAGCTTTTTG
MSN	11	GCATTGACGAATTTGAG TCTA	CCGGGCATTGACGAATTTGAGTCTACTCGAGTAGACTCAAATT CGTCAATGCTTTTTG
MSN	12	GCGGATTAACAAGCGG ATCTT	CCGGGCGGATTAACAAGCGGATCTTCTCGAGAAGATCCGCTT GTTAATCCGCTTTTTG

We then created 15 PDO cell lines, each infected with a lentiviral vector targeting one of the ERM proteins and then we assessed the silencing outcome in each one of the lines through qPCR. From these experiments we selected clones 60, 61, 35, 37, 12 and 11 as the most efficient ones. We then created new PDO lines expressing sequences 61+37+12 or 60+35+11. qPCR analysis of ERM expression in the 61+37+12 indicated an efficient silencing of Ezrin but overexpression of Radixin and Moesin (Fig. R1a). In contrast, combination 60+35+11 resulted in downregulation of Radixin and Moesin but not of Ezrin (Fig. R1b).

[Redacted]

Figure R1: Relative mRNA expression levels of ERM proteins for unsorted PDO cells infected with different combinations of shRNA vectors targeting *Ezr* (61,60), *Rdx* (35,37) or *Msn* (11, 12). Data are represented as the mean \pm s.d.

These unsatisfactory results prompted us to try a new strategy, by combining two *Ezr* targeting sequences (61+60) with clone 11 and 35. This new approach led to an almost complete silencing of Radixin and Moesin and a mild downregulation of Ezrin as indicated in Fig. R1c. Given the difficulties of obtaining large silencing of the three proteins, we selected this PDO line for mechanical analysis. Despite the only partial downregulation of ERM, the LGR5⁻ cells of this PDO line displayed a significant reversion of the observed mechanical phenotype, both in single cell morphology and cluster migratory behavior. These results are consistent with our data based on the linker expression, and further support the role of ERM proteins in explaining the observed mechanical phenotypes. These results are shown in Supplementary Fig. 5e, f, i-k and described in the text on page 9, lines 281-288.

The reviewer also notes that “*the fact that LGR5⁺ cells are better at traversing endothelia can hardly be correlated with the observation that LGR5⁺ cells are better at growing a metastasis AFTER they disseminate (as LGR5⁻ cells) into the secondary organ. So, the proposed model lacks of any real functional validation with cancer assays where the relevance of LGR5 has been established [...].*” While we acknowledge the limitations of *in vitro* when it comes to establishing physiological relevance, we respectfully disagree that our data are in conflict with cancer assays *in vivo*, and particularly with Fumagalli *et al* (Fumagalli et al., 2020). In that study, the authors showed that most metastases are seeded by LGR5⁻ cells, as the reviewer points out. However, when they injected LGR5⁺ and LGR5⁻ cells in equal amounts in mice, they found that LGR5⁺ were more efficient than LGR5⁻ at seeding metastases (Fig 3K and 3L, reproduced below). Thus, according to Fumagalli *et al*, most metastases in animal models are seeded by LGR5⁻ because

most cells escaping the primary tumor are LGR5-, not because LGR5- cells are more efficient than LGR5+ at adhering to or crossing the endothelium. Our data are in line with these results: LGR5- are better at migrating, which might help them escape the primary tumor, but LGR5+ are better at crossing the endothelium, which might help them seed metastases. This point is emphasized in the discussion.

[Redacted]

Figure R2. Reproduction of Fig. 3K and 3L from Fumagalli et al showing that when LGR5+ and LGR5- cells are seeded in equal amounts, LGR5+ cells are more efficient at seeding metastases.

Finally, we agree with the referee that linking our mechanical findings *in vitro* with functional assays *in vivo* would be an impactful extension of our manuscript. A quantitative mechanical characterization of LGR5- and LGR5+ cells *in vivo* is unfortunately not available with current technologies. However, we studied whether the RNA expression patterns of LGR5- and LGR5+ cells in our PDOs capture the gene signatures of populations that have been identified to have a particular role in CRC. To address this point, we performed a bulk RNAseq of the LGR5-, LGR5^{med} and LGR5+ populations. Differential expression analysis revealed that the LGR5+ population was enriched in gene signatures corresponding to ISCs, proliferation, biosynthesis and resistance to chemotherapy. Conversely, LGR5- cells showed enrichment in gene signatures associated to intestinal differentiation (enterocyte, goblet, tuft and mucus-secreting cells) as well as fetal state signatures. LGR5- cells also showed an upregulation of markers for high relapse cells (HRCs) (Cañellas-Socias et al., 2022), including the expression of *Emp1* (encoding epithelial membrane protein 1). Thus, although a functional validation *in vivo* of the metastatic potential of the LGR5+ and LGR5- cells is out of the scope of the current study, these transcriptomic results combined with our *in vitro* findings, support our hypothesis that LGR5- and LGR5+ cells display different phenotypes that are suitable for different functions during CRC metastasis formation. These results are shown in Supplementary Fig. 1d, e and described in the text on page 4-5, lines 122-134.

Reviewer #2 (Remarks to the Author):

In this paper Conti et al. studied the mechanical properties of the different pool of cancer stem cells in colorectal cancer tumors by using colorectal cancer patient derived organoids to

understand if and how these differences contribute to metastatic growth. They found that LGR5+ and LGR5- cells display diverse mechanical properties with LGR5+ cells being stiffer, better adherent to ECM and slower compared to the LGR5- population both when isolated as single cells or in clusters.

They also linked these differences to the downregulation of the ERM proteins, a characteristic that seems conserved throughout colorectal cancer tumors. The study is interesting and tended to approach a recurrent problem in CRC which is the metastatic relapse even after removal of the primary tumor. How this occurs from LGR5- cells and which mechanical properties are functionally relevant for the metastatic cascade is still not well known. However, the study remains superficial regarding some aspects and points out mainly a correlation of LGR5 expression and mechanical properties while the authors did not untangle the relationship between the mechanical and biochemical properties to understand if mechanical adaptability is the main driver of the metastatic progression. Within the study three main areas of concerns are noticed:

We thank the referee for their thorough analysis of our work and for their constructive feedback. Our point-by-point response to their comments is as follows.

Major points:

(1) The authors used a previously published CRC patient derived organoid line and sorted the cells in LGR5-, LGR5^{med} and LGR5+ cells based on the LGR5 intensities. However, the study lacks a detailed analysis of the cell type identity and cell heterogeneity of the different populations. For example, the authors cite the paper Ganesh et al., where they show that the L1CAM gene is required for metastatic colonization and its expression overlap with LGR5+ cells. However, the authors didn't do any deep characterization of the three population they used in the study. This becomes especially evident for the LGR5- population that can be divided in two distinct group even before sorting. It is not clear whether this population comprises different cell states or even different cell types. Indeed, when looking at the cluster experiments after 15h a LGR5+ cell appear. Is this a rare event and how does this influence the mechanical properties of the cluster?

We thank the referee for raising this point, which was missing in the original manuscript. Following this comment and with the objective of achieving a more comprehensive understanding of the molecular features of the three populations, we performed a bulk RNAseq analysis of the LGR5-, LGR5^{med} and LGR5+ populations. This analysis revealed that the LGR5+ population was enriched in gene signatures corresponding to ISCs, proliferation, biosynthesis and resistance to chemotherapy. Conversely, LGR5- cells showed enrichment in gene signatures associated to intestinal differentiation (enterocyte, goblet, tuft and mucus-secreting cells) as well as fetal state signatures. LGR5- cells also showed an upregulation of markers for high relapse cells (HRCs), including the expression of *Emp1* (encoding epithelial membrane protein 1) (Cañellas-Socias et al., 2022). Thus, the 3 distinct cell populations in our study reflect the known features of CRC hierarchy. These results are shown in Supplementary Fig. 1d, e and described in the text on pages 4-5, lines 122-134.

As for *L1cam*, we found no significant differences in gene expression between the three populations. As discussed below, we did not find differences in *L1cam* between LGR5+ and LGR5- in the patient cohort either. Our findings are consistent with those of Ganesh, Massagué and colleagues who showed that L1CAM is expressed in both Lgr5+ and Lgr5- cells in PDOs (Ganesh et al., 2020).

[Redacted]

Figure R3. *L1cam* normalized expression in LGR5-, LGR5^{med} and LGR5+ cells as quantified by Bulk RNA-seq. Statistical analysis was performed using t-tests.

(2) Regarding the migration assay: it is known that migration is a collective property and indeed clusters move faster than single cells. Yet, it remains unclear how cells in the cluster move. A more detailed study of how cells move individually under confinement and how they move as a cluster would benefit the study. Within a migrating cluster, do all the cells show the same polarity? Do they move all together in the same direction? What mode of migration do they follow? The authors could further analyze the actin cytoskeleton and myosin and see if this can be the main difference between positive and negative clusters. Can migration be inhibited or equalized between the two populations using inhibitors of the iMC linker?

We agree that further characterization of the migration mode of the clusters was in order. Unlike for single cells, the mechanisms of cluster migration remain poorly understood. Recent studies from our lab provide a new physical framework for cluster migration based on the so-called physics of active wetting (Pérez-González et al., 2019; Pallarès et al., 2022). Inspired by how a fluid droplet wets a surface, this theoretical and experimental framework shows that cell clusters (considered as active contractile droplets) spread and migrate on their substrate depending on the physical balance between their traction force, contractility and surface tension. A non-trivial prediction of this biophysical picture is that clusters with either high or low spreading (i.e., high or low wettability, respectively) display low migration velocity, whereas clusters with intermediate spreading (i.e., neutral wetting) display optimal velocity. To address whether the migration of PDO clusters can be understood under the mechanisms of active wetting, we measured the contact angle between the clusters and their substrate. We found that LGR5^{low}, LGR5^{med}, and LGR5^{high} display progressively decreasing contact angles, consistent with their increasing spreading. Moreover, we found that the contact angle of the fastest clusters (LGR5^{low}) is $\sim 100^\circ$, which falls within the optimal velocity regime (neutral wetting) predicted by theory and found experimentally in Pallarès et al (Pallarès et al., 2022).

The higher velocity in the neutral wetting regime found for LGR5^{low} vs the lower velocity in the high wetting regime found for LGR5^{high} can be explained by two main mechanisms. The first one is increased surface tension due to the overexpression of ERMs in LGR5^{low} clusters. Consistent with this possibility, we found that LGR5^{high} clusters expressing the iMC-linker are less spread and faster than control LGR5^{high} clusters, whereas LGR5^{low} clusters in which ERMs are partially knocked down are slower and more spread than both control LGR5^{high} and LGR5^{low} clusters. The second mechanism would be an increase in contractility in LGR5^{low} clusters. To test this possibility, and as suggested by the referee, we measured levels of phosphorylated myosin light chain (pMLC) of the cluster most basal plane. We found lower pMLC in LGR5^{low}, inconsistent with tilting the force balance towards a neutral wetting regime observed in our experiments. Taken together, our findings indicate that LGR5^{low} clusters display higher motility than LGR5^{med} and LGR5^{high} because their increased cell to cortex attachment (rather than pMLC levels) promotes a state of neutral active wetting regime that optimizes motility. We present our new measurements of contact angle in Fig 3g-i, pMLC quantification in Supplementary Fig. 2a-c and discuss them on page 7-8, lines 219-238.

(3) The authors identify membrane to cortex attachment via ERM as important mediator of the different mechanical properties observed in LGR5+ and LGR5- cells. This is one of the key findings of this study and a better characterization of the mechanism by which ERM protein induces a change in mechanical phenotype would be important also in respect to the in vivo data. It is not clear for example if the mechanism is mediated by the difference in the phosphorylation status of ERM and their activation and one way to check this would be using chemical or genetic perturbations. This can be also used to understand if it is possible also to convert the mechanical phenotype of the LGR5- cells towards the LGR5+ cells. Moreover, the authors should check the total level of ERM together with their phosphorylation status. As the authors define ERM as the main driver of the mechanical properties of LGR5- and + populations, leading to dissemination or extravasation of the tumor, it would be important to check if the LGR5- phenotype, induced by the expression of iMC in LGR5+ cells, have further consequences like changes in cell identity, proliferation and/or migration.

To better characterize the molecular mechanism underlying the uncovered mechanical phenotypes we analyzed the levels of total ERM showing that their upregulation is not only at the transcription level (as shown by the qPCR results) and at the phosphorylation level (as shown by western blotting of pERM) but also at the total protein level. Furthermore, we created a stable PDO line expressing shRNA targeting Ezrin, Radixin and Moesin, which showed an almost total downregulation of Radixin and Moesin and a mild downregulation of Ezrin (see response to referee 1 and Figures R1 and R2 above). The LGR5- cells of the new PDO line displayed a significant reversion of the observed mechanical phenotype, both in single cell morphology and cluster migratory behavior (Fig. 5e, f, i-k; in the text these results are described on page 9, lines 281-288).

Regarding the final question in this section, which concerns the impact of iMC-linker expression on cell identity, proliferation, and migration, we performed the following experiments. To investigate the effects of the linker expression on migration we tracked LGR5+ clusters seeded on 3kPa gels coated with Collagen I (same conditions we used for all the cluster experiments). LGR5^{high} clusters expressing the linker were significantly faster compared to LGR5^{high} clusters not expressing the linker, suggesting that a high MCA enhances CRC collective motility. These results

are shown in Fig. 5j, k and described in the text on page 10, lines 303-309. To explore the effects of iMClinker expression on cell identity and proliferation we induced it by doxycycline treatment in PDOs cultured in BME for 48 and 72h. We then assessed proliferation through flow cytometry cell cycle analysis based on the measurement of DNA content. We also measured Tdtomato fluorescence to estimate the amount of LGR5+ cells in the PDO population. Expression of the linker in the PDOs resulted in a reduction of the percentage of LGR5+ proliferative cells coupled with a surprising increase in the fraction of LGR5+ cells in the population (Fig. R4). These results indicate that increased membrane to cortex attachment is a key phenotypic feature of LGR5- but one that promotes LGR5+ expression, thus pointing to non-trivial regulatory layers to sustain the LGR5+ and LGR5- populations. These results are intriguing, and we feel they deserve a deeper understanding before being published. We show them below for the benefit of the reviewers and leave their detailed understanding for further work.

[Redacted]

Figure R4: Percentages of LGR5+ cells and LGR5+ proliferative cells following Doxycycline induced iMC-linker expression. Doxycycline treatment was either for 48 or 72 h. Proliferation was assessed through DNA based cell cycle analysis by flow cytometry.

(4) Very interesting is also the ERM transcriptional difference the authors find in the patient cohort. However, to strengthen the relevance of the finding a more extensive analysis of the transcriptomic data would be important. How is the expression of ERM in healthy tissue and is the difference between ERM expression a cancer specific trait that is acquired or exists also in healthy cells?

We thank the referee for their comment which allowed us to obtain a more comprehensive understanding of the molecular mechanisms and their link to molecular features of healthy tissue. As mentioned briefly in our original discussion, ERM proteins have been previously reported to be expressed in the differentiated healthy tissue (refs 75-77). To expand on this knowledge, and as suggested by the reviewer, we analyzed scRNAseq data of healthy tissue from 5 donors. This analysis confirms that ERM protein expression and in particular Ezrin expression, is not an exclusive feature of CRC. Indeed, Ezrin is expressed in most of the normal colon epithelial cells,

especially colonocytes and Bestrophin 4 cells (BEST4+). Radixin was found to be mostly expressed by enteroendocrine cells (EECs) while Moesin had an overall low expression and is mostly expressed by Tuft cells and BEST2+ Goblet cells. These results are shown in Fig. 6e-g and described in the text on page 11, lines 349-355.

Do other proteins interacting with ERM show a similar expression pattern?

In an open, unmasked conformation the ERM C-terminal domain binds to actin filaments while the N-terminal FERM domain can bind a vast array of transmembrane proteins including CD44, SPN, ICAM2, 3, VCAM1, syndecan-2 etc. To perform the transcriptomic analysis of the SMC cohort, we selected some of the ERM interacting proteins (based on a list extracted from (McClatchey, 2012), on the basis of functional relevance.

ERM interacting protein (direct association)	Function
Integrin β2 (ITGB2)	Interaction with ECM
CD44	Proliferation, cell differentiation, cell migration, angiogenesis, presentation of cytokines, chemokines, and growth factors, and docking of proteases at the cell membrane, signaling for cell survival.
CD43 (SPN)	Immune function
ICAM1/2/3/VCAM-1	Cell surface glycoprotein, typically expressed on endothelial cells and cells of the immune system.
Layilin (LAYN)	Membrane-binding partner for talin, recruited in peripheral ruffles of spreading cells
Syndecan-2 (SDC2)	Functions as an integral membrane protein and participates in cell proliferation, cell migration and cell-matrix interactions via its receptor for extracellular matrix proteins.

We found no significant differences in the expression of ERM interacting proteins CD44, SPN (CD43) and ICAM1, 2, 3, LAYN, SDC2 between the LGR5- and LGR5+ cells in the SMC cohort (Fig. R5)

How different is the expression of L1CAM?

We found no differences also in *L1cam* expression (Fig. R5).

[Redacted]

Figure R5: Gene expression levels of L1CAM, CD44, SPN, ICAM1/2/3, LAYN, SDC2, ITGB2 in epithelial tumor cells from CRC patients in the SMC cohort summarized by patient through the average. Each dot corresponds to the average expression levels of one patient. summarized by patient through the average. The boxes center line represents the median. The box limits represent the first and third quartiles. Whiskers indicate maximum and minimum values. n = 15. A linear model was fitted to the data to assess significance.

Why differently to the PDOs, only Ezrin upregulated in the negative LGR5 population?

In our original manuscript we showed a tendency of Ezrin to be upregulated in the LGR5- population but this result did not reach statistical significance. Given the importance of this result we carried out additional qPCR experiments, which showed Ezrin is overexpressed in the LGR5- cells both in the PDOs and in the patient cohort. We confirmed this result using bulk RNAseq. The updated plots are shown in Fig. 5b and described on page 9 lines 272-275.

Within the LGR5- populations is ERM differently expressed in different cell types or differently expressed in cells that are more or less differentiated?

To address this point, we investigated the content of LGR5+ and KRT20+ cells in each patient of the cohort, finding high heterogeneity in the differentiation degree of tumors between patients. We then analyzed ERM expression as a function of LGR5 and KRT20 expression (Supplementary Fig. 5). We found that Ezrin expression was highest in the KRT20 cells, supporting that similarly to the normal intestine, Ezrin expression is linked to cell differentiation in CRC. As for Radixin and Moesin, this link was not obvious, with KRT20+ cells expressing the highest levels of these two genes only in a small subset of patients. These results suggest that while Ezrin expression may originate from an intestinal differentiation program conserved in CRC, Radixin and Moesin expression are not necessarily coupled with tumor differentiation.

Minor Points:

a. In CRC as well as in PDOs, CK20 and LGR5 are expected to have complementary expression domains. Yet, Fig1a shows co-staining of CK20 and Lgr5. Can the authors comment on this?

The PDO model used in the study was engineered by knock-in CRISPR/Cas9-mediated homologous recombination where an IRES-TdTomato-WPRE-BGHpA cassette was inserted after the stop codon of the *Lgr5* gene, thereby fluorescently labelling the LGR5+ cells using the LGR5 endogenous promoter as a driver of TdTomato expression. Therefore, the Tdtomato labelling is a direct readout of LGR5 expression in the PDO cells. The coexistence in a few cells of TdTomato and CK20, might indicate a transition state or a delay caused by the degradation time of the TdTomato compared to the real-time expression of LGR5. To further establish the validity of our model (see (Cortina et al., 2017) for extensive characterization of the organoids), we quantified the expression of CK20 and Tdtomato in 3D organoids embedded in BME. As shown in Supplementary Fig. 1a, the expression patterns of Tdtomato and CK20 are as expected, with cells expressing high levels of CK20, displaying low levels of LGR5 expression and cells expressing high levels of Tdtomato displaying low levels of CK20.

b. Can the authors comment on why the stiffnesses of the PAA gels were not kept equal throughout the entire study? In some experiments the stiffness was 0.3, 3.5 and 11 kPa and in other 0.3, 1.5, 5 and 11kPa. What is the reasoning for the choice of this range of stiffness and how relevant is this stiffness in the vivo context? Moreover, the clusters analyses were all done on a substrate of 1kPa, and the authors never show if there is any difference in shape of migration at difference stiffness. Since they show emergent mechanical properties only in multicellular clusters, maybe also the response to stiffness is different compared to single cells.

The substrate rigidities used for single cell morphology, traction and velocity experiments were 0.5, 1.5, 5 and 11 while for YAP staining we used 0.5, 3 and 30kPa. The reason for this

discrepancy is mainly technical. Our goal was to use a broad rigidity range compatible with our techniques and with the *in vivo* spectrum of CRC. Our cells were unable to deform gels with a stiffness >11 kPa, which forced us to restrict our traction microscopy experiments to this rigidity. This limitation did not apply to YAP stainings, which allowed us to expand the rigidity range to 30kPa gels.

To address the question of how stiffness may affect cluster migration, we seeded PDO clusters on 11kPa gels coated with Collagen I. We found a similar pattern to the one observed at 3kPa (the substrate rigidity that was used for all cluster experiments), with the LGR5^{low} clusters moving faster than the LGR5^{med} and LGR5^{high} clusters (Supplementary Fig. 2a, b; page 7 lines 219). Thus, we conclude that the stemness-dependent differences in migratory behavior of PDO clusters is not a feature of a specific rigidity.

c. As single cells LGR5- and Lgr5+ cells differ in nuclear YAP. Is this difference also present in the 3D organoids? Are differences only on the localization of YAP or also its expression level and what functional implications does this have?

To address this question, we performed YAP immunostainings in LGR5-Tdtomato PDOs cultured in 3D. Interestingly, we observe similar levels of YAP expression in all the cells with a prevailing localization in the cytosol in an LGR5-independent manner (Fig. R6). This stable cytosolic localization is likely a result of a) the extremely low stiffness of the matrix surrounding the organoids, b) the observation that all cells have a packed with a similar shape regardless of LGR5 expression. These mechanical conditions appear insufficient to activate the mechanobiological pathways that allow the translocation of YAP into the nucleus, and hence no differences are observed in YAP localization as a function of LGR5 expression.

However, this does not imply that our YAP results on 2D substrates lack functional implications *in vivo*. Extensive documentation indicates that in tumors are much stiffer than the ECM in which PDOs are cultured in 3D (Bauer et al., 2020; Evans et al., 2012; Rouvière et al., 2017; Shahryari et al., 2019), which likely brings these organoids into a different regime than that of 3D culture. In addition, the contractility of CAFs is also known to contribute to YAP localization (Barbazan et al., 2021), now accepted in Nat Comms). Further research is necessary to comprehend whether mechanics, particularly through YAP, exerts distinct effects on different cell types within a tumor in its physiological context.

[Redacted]

Figure R6. Confocal images of Lgr5-Tdtomato PDOs immunostained for YAP. In parallel, these PDOs were stained with phalloidin and Hoechst. In the panel at the right, we show LGR5 mean levels and YAP nuclear-to-cytosolic ratio of 284 cells of n=6 organoids. The Pearson correlation coefficient $r = -0.018$, which indicates a very low correlation between both variables. Bar, 20 μm .

d. From FigureS2b it is not clear if in the confinement experiment the authors seeded a mixed population in the cell confiner. It would be interesting to know the % of total population at different confinement for the three different population.

In the confinement experiments the cells were seeded at a 1:1 ratio. Only the LGR5- and LGR5+ cells were included in these experiments. This is clarified in the methods section.

e. Shape analysis measurement on the cell clusters were done on the entire cluster and not for each cell inside the cluster. It would be interesting to analyze cell shape in their initial 3D environment (in the PDO2) to check if the shape features are preserved after dispase treatment or sorting.

Although it is an interesting point, addressing it would require overcoming substantial image analysis challenges and it is out of the scope of this study. Nevertheless, we thank the referee for this suggestion and will consider following it in future studies.

f. Regarding the style of the figures, I would suggest that the authors combine Fig 5,6 and 7 in two figures.

We agree and changed the figures accordingly.

References

- Barbazan, J., Pérez-González, C., Gómez-González, M., Dedenon, M., Richon, S., Latorre, E., Serra, M., Mariani, P., Descroix, S., Sens, P., Trepas, X., Vignjevic, D.M., 2021. Cancer-associated fibroblasts actively compress cancer cells and modulate mechanotransduction. <https://doi.org/10.1101/2021.04.05.438443>
- Bauer, J., Emon, M.A.B., Staudacher, J.J., Thomas, A.L., Zessner-Spitzenberg, J., Mancinelli, G., Krett, N., Saif, M.T., Jung, B., 2020. Increased stiffness of the tumor microenvironment in colon cancer stimulates cancer associated fibroblast-mediated prometastatic activin A signaling. *Sci. Rep.* 10, 50. <https://doi.org/10.1038/s41598-019-55687-6>
- Cañellas-Socias, A., Cortina, C., Hernando-Momblona, X., Palomo-Ponce, S., Mulholland, E.J., Turon, G., Mateo, L., Conti, S., Roman, O., Sevillano, M., Slebe, F., Stork, D., Caballé-Mestres, A., Berenguer-Llergo, A., Álvarez-Varela, A., Fenderico, N., Novellademunt, L., Jiménez-Gracia, L., Sipka, T., Bardia, L., Lorden, P., Colombelli, J., Heyn, H., Trepas, X., Tejpar, S., Sancho, E., Tauriello, D.V.F., Leedham, S., Attolini, C.S.-O., Batlle, E., 2022. Metastatic recurrence in colorectal cancer arises from residual EMP1+ cells. *Nature* 611, 603–613. <https://doi.org/10.1038/s41586-022-05402-9>
- Cortina, C., Turon, G., Stork, D., Hernando-Momblona, X., Sevillano, M., Aguilera, M., Tosi, S., Merlos-Suárez, A., Stephan-Otto Attolini, C., Sancho, E., Batlle, E., 2017. A genome editing approach to study cancer stem cells in human tumors. *EMBO Mol. Med.* 9, 869–879. <https://doi.org/10.15252/emmm.201707550>
- Elosegui-Artola, A., Oria, R., Chen, Y., Kosmalska, A., Pérez-González, C., Castro, N., Zhu, C., Trepas, X., Roca-Cusachs, P., 2016. Mechanical regulation of a molecular clutch defines force transmission and transduction in response to matrix rigidity. *Nat. Cell Biol.* 18, 540–548. <https://doi.org/10.1038/ncb3336>

- Evans, A., Whelehan, P., Thomson, K., Brauer, K., Jordan, L., Purdie, C., McLean, D., Baker, L., Vinnicombe, S., Thompson, A., 2012. Differentiating benign from malignant solid breast masses: value of shear wave elastography according to lesion stiffness combined with greyscale ultrasound according to BI-RADS classification. *Br. J. Cancer* 107, 224–229. <https://doi.org/10.1038/bjc.2012.253>
- Fumagalli, A., Oost, K.C., Kester, L., Morgner, J., Bornes, L., Bruens, L., Spaargaren, L., Azkanaz, M., Schelfhorst, T., Beerling, E., Heinz, M.C., Postrach, D., Seinstra, D., Sieuwerts, A.M., Martens, J.W.M., van der Elst, S., van Baalen, M., Bhowmick, D., Vrisekoop, N., Ellenbroek, S.I.J., Suijkerbuijk, S.J.E., Snippert, H.J., van Rheenen, J., 2020. Plasticity of Lgr5-Negative Cancer Cells Drives Metastasis in Colorectal Cancer. *Cell Stem Cell* 26, 569-578.e7. <https://doi.org/10.1016/j.stem.2020.02.008>
- Ganesh, K., Basnet, H., Kaygusuz, Y., Laughney, A.M., He, L., Sharma, R., O'Rourke, K.P., Reuter, V.P., Huang, Y.-H., Turkekul, M., Er, E.E., Masilionis, I., Manova-Todorova, K., Weiser, M.R., Saltz, L.B., Garcia-Aguilar, J., Koche, R., Lowe, S.W., Pe'er, D., Shia, J., Massagué, J., 2020. L1CAM defines the regenerative origin of metastasis-initiating cells in colorectal cancer. *Nat. Cancer* 1, 28–45. <https://doi.org/10.1038/s43018-019-0006-x>
- McClatchey, A.I., 2012. ERM proteins. *Curr. Biol.* 22, R784–R785. <https://doi.org/10.1016/j.cub.2012.07.057>
- Pallarès, M.E., Pi-Jaumà, I., Fortunato, I.C., Grazu, V., Gómez-González, M., Roca-Cusachs, P., de la Fuente, J.M., Alert, R., Sunyer, R., Casademunt, J., Trepát, X., 2022. Stiffness-dependent active wetting enables optimal collective cell durotaxis. *Nat. Phys.* 1–11. <https://doi.org/10.1038/s41567-022-01835-1>
- Pérez-González, C., Alert, R., Blanch-Mercader, C., Gómez-González, M., Kolodziej, T., Bazellieres, E., Casademunt, J., Trepát, X., 2019. Active wetting of epithelial tissues. *Nat. Phys.* 15, 79–88. <https://doi.org/10.1038/s41567-018-0279-5>
- Rouvière, O., Melodelima, C., Hoang Dinh, A., Bratan, F., Pagnoux, G., Sanzalone, T., Crouzet, S., Colombel, M., Mège-Lechevallier, F., Souchon, R., 2017. Stiffness of benign and malignant prostate tissue measured by shear-wave elastography: a preliminary study. *Eur. Radiol.* 27, 1858–1866. <https://doi.org/10.1007/s00330-016-4534-9>
- Shahyari, M., Tzschätzsch, H., Guo, J., Marticorena Garcia, S.R., Böning, G., Fehrenbach, U., Stencel, L., Asbach, P., Hamm, B., Käs, J.A., Braun, J., Denecke, T., Sack, I., 2019. Tomoelastography Distinguishes Noninvasively between Benign and Malignant Liver Lesions. *Cancer Res.* 79, 5704–5710. <https://doi.org/10.1158/0008-5472.CAN-19-2150>

REVIEWERS' COMMENTS

Reviewer #1 (Remarks to the Author):

The authors performed experiments along my previous requests, and now enriched their data by better exploring the functional link between ERM proteins and CRC mechanics with loss-of-function reagents. Some of the differences in key mechanical properties between LGR5+ and LGR5- cells are also observed in another patient-derived CRC population. The functional relevance of these mechanical differences remains unknown in an in vivo metastasis setting. Therefore, I would recommend that the authors clearly state in the abstract and discussion how this study provides an in vitro model of in vivo phenotypes observed by others, without however establishing this link experimentally.

Reviewer #2 (Remarks to the Author):

In the revised manuscript, Conti et al. have significantly expanded their work and answered to all the major and minor points addressed. The updated text is well written and has improved interpretability of the data, leading to a clearer message. Indeed, the revised manuscript now demonstrates more robustly that differences of mechanical properties in Lgr5+ and Lgr5- populations are linked to ERM proteins and provided a deeper analysis of the in vivo data. Further, the authors have effectively addressed concerns related to the identity of the different populations of the CRC PDOs and provided a clearer and more comprehensive discussion on wetting as biophysical framework for experimental observations in cluster migration.

However, while major concerns were addressed by the authors, there are still some minor points to be addressed. Once the authors reply to these last minor concerns, the paper might be considered to be published without any further revision.

Please find below the list of the minor points to be addressed from the authors:

- 1) The bulk RNA-seq analysis of the different LGR5 populations revealed some more insights into the transcriptional differences. However, it is not very clear how the LGR5med population differ from the other two, even though they display different mechanical properties. Can the authors comment on that?
- 2) For supplementary figure 1d and 1e, include a clear description in the graph explaining the meaning of the color scale. Additionally, use the full term "normalized enrichment score (NES)" at least once, rather than relying solely on the abbreviation. The current combination of sample comparisons and enrichment scores in the graph could lead to confusion; if feasible, consider adjusting the graph layout for clarity. Please provide count tables for the bulk sequencing if possible and provide lists or proper references for the gene-sets in the method section.
- 3) The authors checked the total level of ERM together with their phosphorylation status by western blot, however a quantification of the result would make clearer that also the total ERM protein goes up in the LGR5- population. As now in Fig. 5c it is not very clear.
- 4) In Figure 5e please incorporate the error bars and conduct a significance test.
- 5) Line 145-147 the authors refer to a figure that does not correspond to the text. Maybe they meant to relate to Fig. 1g-i? Please clarify this.
- 6) Please include quantifications and images for Lgr5- clusters in 5j and k as controls.
- 7) In the text, line 286-288, the authors state that the downregulation of the ERM leads to a Lgr5+ cluster exhibiting a Lgr5+ cluster phenotype. This appears to be a typographical error

and likely refers to the Lgr5⁻ cluster resembling the Lgr5⁺ cluster. Please make the necessary correction.

8) How was calculated the fluorescence intensity of LGR5? FigS1b and main figures (as Fig3c) show very different orders of magnitudes.